## OPEN

# Structure of a thylakoid-anchored contractile injection system in multicellular cyanobacteria

Gregor L. Weiss[1,4], Fabian Eisenstein [1,3,4], Ann-Katrin Kieninger[2], Jingwei Xu [1], Hannah A. Minas [1], Milena Gerber[1], Miki Feldmüller[1], Iris Maldener[2], Karl Forchhammer [2] and Martin Pilhofer [1]✉

Contractile injection systems (CISs) mediate cell–cell interactions by phage tail-like structures, using two distinct modes of action: extracellular CISs are released into the medium, while type 6 secretion systems (T6SSs) are attached to the cytoplasmic membrane and function upon cell–cell contact. Here, we characterized a CIS in the multicellular cyanobacterium *Anabaena*, with features distinct from extracellular CISs and T6SSs. Cryo-electron tomography of focused ion beam-milled cells revealed that CISs were anchored in thylakoid membrane stacks, facing the cell periphery. Single particle cryo-electron microscopy showed that this unique in situ localization was mediated by extensions of tail fibre and baseplate components. On stress, cyanobacteria induced the formation of ghost cells, presenting thylakoid-anchored CISs to the environment. Functional assays suggest that these CISs may mediate ghost cell formation and/or interactions of ghost cells with other organisms. Collectively, these data provide a framework for understanding the evolutionary re-engineering of CISs and potential roles of these CISs in cyanobacterial programmed cell death.

Bacteria evolved a variety of sophisticated macromolecular assemblies to translocate effectors into extracellular space or directly into target cells[1,2]. Contractile injection systems (CISs) are a diverse, yet evolutionarily related, group of assemblies that mediate bacterial cell–cell interactions[3–6]. Their contractile apparatus is homologous to contractile phage tails, using a contractile outer sheath to propel an effector-loaded tube into target cells. One end of the sheath–tube module is attached to a baseplate, which triggers sheath contraction by a conformational change[7].

Depending on how the baseplate is anchored to a membrane before firing, two fundamentally different modes of action emerge: type 6 secretion systems (T6SSs) are anchored to the cytoplasmic membrane and act in a cell–cell contact-dependent manner, stabbing the inner tube through the own cell envelope and into the adjacent neighbouring cell[8,9]. By contrast, extracellular CISs (eCIS) are released into extracellular space, where they can bind to the surface of a target cell via tail fibres and puncture its envelope[3–5].

Several phylogenetically closely related CISs have recently been studied and are particularly intriguing. This group contains representatives of both eCISs and T6SSs. Characterized representatives are insecticidal eCISs such as antifeeding prophages (AFPs) and *Photorhabdus* virulence cassettes (PVCs), as well as large arrays of eCISs that induce the metamorphosis of tubeworm larvae (MACs)[10–14]. Another member of this group, found in a symbiont of amoebae, assembles bundles of T6SSs that mediate interactions with host membranes[15]. Importantly, recent bioinformatic searches detected related gene clusters in hundreds of diverse bacterial and even archaeal genomes[11,15–17].

Here, we study a cyanobacterial CIS gene cluster using an integrative approach. Cyanobacteria are among the most abundant organisms on Earth and play key roles in geochemical cycles[18]. While 40% of sequenced cyanobacterial genomes contain at least one CIS gene cluster[16], their structure, mode of action and role in the cyanobacterial lifecycle remain unknown.

## Results and discussion

***Anabaena* CISs are anchored in thylakoid membrane stacks.** We set out to study cyanobacterial CISs in the multicellular model organism *Anabaena* sp. PCC 7120 (hereafter *Anabaena*). The previously identified CIS gene cluster predicts 15 open reading frames (*all3313–all3327*)[16]. We relabelled the genes with the prefix '*cis*' and numbered them according to their homology to the close eCIS relative AFP (Fig. 1a and Supplementary Table 1). Since no homologue for *all3314* was identified in AFP, we labelled it *cis19*, expanding the existing numbering. In addition to the structural components, we detected two putative effector genes (*alr3312* and *alr3311*). To test whether the gene cluster is expressed and produces CIS-like assemblies, we prepared a crude purification of CISs. Negative-stain electron microscopy (EM) images revealed numerous CIS-like assemblies resembling eCISs in extended and contracted conformations (Supplementary Fig. 1a). Mass spectrometry (MS) analysis of the purification detected 13 out of 17 proteins encoded in the gene cluster (asterisks in Fig. 1a and Supplementary Fig. 1b). To localize CIS proteins, we generated antibodies against tube and sheath (Cis1/2) and detected both almost exclusively in the cell pellet and not in the culture supernatant (Supplementary Fig. 1c). This result was inconsistent with an eCIS mode of action. To further locate CISs within *Anabaena* filaments, we constructed a C-terminal fusion of a putative baseplate component (Cis11) with super-folder green fluorescent protein (sfGFP). Fluorescence light microscopy (fLM) revealed foci in every analysed cell, with an average of ~18 foci per cell (n = 129 cells; Supplementary Fig. 2). Foci were predominantly located in the cell periphery (Fig. 1b and Supplementary Fig. 3).

Next, we determined the in situ structure of *Anabaena* CIS by plunge freezing cells on EM grids, thinning them by cryo-focused

[1]Department of Biology, Institute of Molecular Biology & Biophysics, Eidgenössische Technische Hochschule Zürich, Zurich, Switzerland. [2]Interfaculty Institute of Microbiology and Infection Medicine Tübingen, Organismic Interactions, University of Tübingen, Tübingen, Germany. [3]Present address: Graduate School of Medicine, University of Tokyo, Tokyo, Japan. [4]These authors contributed equally: G. L. Weiss, F. Eisenstein. ✉e-mail: pilhofer@biol.ethz.ch

ion beam (cryoFIB) milling and imaging them by cryo-electron tomography (cryoET)[19–21]. Strikingly, CISs were always found in the extended conformation and always anchored within a pore in the thylakoid membrane stacks (TMs) (Fig. 1c–f). TMs are cyanobacterial intracellular photosynthetic membranes. Most often, CISs were embedded in the outermost TM (90.4%, $n = 189$ out of 209 CISs, 99 tomograms total) and with the baseplate facing the cell envelope (98.6%, $n = 206$ out of 209; Fig. 1g and Supplementary Fig. 4a). Minor fractions were seen in the second (4.9%, $n = 10$ out of 209), third (3.8%, $n = 8$ out of 209) or fourth (0.9%, $n = 2$ out of 209) TM (Fig. 1g).

To reveal the membrane anchoring mechanism in more detail, we performed subtomogram averaging of 209 CISs. The resulting average revealed a CIS-like assembly (119 nm total length), comprising a sheath–tube module and a large baseplate complex (Supplementary Fig. 4b). Focused alignment of the baseplate region improved the level of detail, revealing a pore within the TM with a diameter of ~35 nm (Fig. 1h and Supplementary Fig. 4c). Interactions between CIS and TM were established by distinct sets of six 'narrow connectors' (10 nm long and 1.5 nm in diameter) and six 'wide connectors' (5 nm long and 2.5 nm in diameter) (Fig. 1h,i). Other unique features included a hexagonal cage encapsulating the spike and a crown-like extension of the baseplate (Fig. 1h,i).

**Atomic model of CIS reveals unique features.** To identify the proteins responsible for the unique structural characteristics of *Anabaena* CIS, we optimized the native purification of the complex and used single particle analysis (SPA) cryoEM to obtain high-resolution structures (Fig. 2 and Supplementary Fig. 5). The resulting maps of the baseplate and of the distal end cap reached resolutions of ~2.9 and ~2.8 Å, respectively (Supplementary Fig. 6). This allowed us to build de novo atomic models of 13 proteins encoded in the CIS gene cluster. Using the dimensions obtained from the in situ subtomogram average (Supplementary Fig. 4b), a full model of the CIS could be assembled (Fig. 2a and Supplementary Video 1). All proteins detected by MS could be identified within the structure, except for Alr3312 and Alr3311, which both contain the typical CIS effector domain DUF4157 (ref. [22]). Our observation of *Anabaena* CISs with both filled and empty tube lumens could suggest that these proteins might be loaded as cargo into the tube lumen similar to an effector of MACs (Supplementary Fig. 7)[23]. Lack of structural rigidity or symmetry could explain the absence of these proteins in the SPA map.

The distal end of the CIS was capped by a tail terminator complex (Extended Data Fig. 1a). In contrast to previously studied CISs (Extended Data Fig. 1b,c), the cap of the *Anabaena* CIS was composed of two proteins, Cis16A and Cis16B, forming a C6-symmetrical complex that terminated the inner tube and sheath,

respectively. The structure of the sheath–tube was highly conserved to related CISs and was composed of 23 hexameric layers of the sheath protein Cis2 and 22 hexameric rings of the inner tube protein Cis1 (Fig. 2a and Extended Data Fig. 2). The transition from the inner tube with C6 symmetry to Cis8, the VgrG-like spike with C3 symmetry, was facilitated by Cis5 and Cis7, which formed the first two layers of the inner tube. The Cis8 spike trimer was tipped with an unresolved density probably attributed to a single copy of the PAAR-like Cis10. The cavity of the spike towards the tube lumen was plugged by three copies of Cis6 (Extended Data Fig. 3). Even though AFP and PVC contain a Cis6 homologue (Afp6/Pvc6), it was not resolved previously[10,11]. Further details regarding this spike plug and the bipartite cap in a related system are discussed in an accompanying study[24].

The highly conserved baseplate component Cis9 linked the lateral baseplate components Cis11 and Cis12 to the first layer of the sheath. Cis11 and Cis12 formed a hexamer of heterodimers surrounding the spike as seen in previously studied CISs (Fig. 2c). When compared to Afp11/Pvc11, Cis11 harboured three extra domains including an extended carboxy terminus (Extended Data Fig. 4). The C-terminal Cis11 extension contacted a trimer of Cis13, a tail fibre-like protein. The second additional Cis11 domain seemed to stabilize interactions with Cis12, whereas the third extra domain composed the cage surrounding the spike (Fig. 2a–c), which was observed in the in situ structure (Fig. 1i). The function of this cage remains unknown and could range from compartmentalization of possible effector molecules to acting as a receptor for target cells[24]. The cage was further surrounded by the crown, consisting of six trimers of Cis19, which are not present in other studied CISs (Fig. 2c,d). The upper domains of Cis19 broke the pseudo-C3 symmetry and extended far beyond the baseplate. These regions, however, could not be modelled due to the high degree of flexibility (Fig. 2e).

**Baseplate and tail fibre extensions mediate TM anchoring.** We next focused on how the CIS established connections with the TM. The wide connector observed in the in situ subtomogram average was composed of an extra domain of the baseplate component Cis12 (Fig. 3a,b). As the periphery of Cis12 could only be partially modelled, it remains unclear whether Cis12 bound the membrane directly or interacted via an unknown thylakoid membrane protein. Secondary structure prediction did not identify any transmembrane domains but detected a putative amphipathic helix within the unresolved part of Cis12 that might facilitate membrane interactions (Fig. 3b and Supplementary Fig. 8a).

The narrow connector was formed by the tail fibre-like Cis13 trimer (Fig. 3a,c). The amino-terminal domains of Cis13 proteins bound the baseplate while the C termini formed a triple alpha helix extending towards the TM and penetrating it. Although the

**Fig. 1 | *Anabaena* CISs are anchored in thylakoid membrane stacks. a**, CIS gene cluster of *Anabaena* sp. PCC 7120. Genes were renamed according to homology to previously studied CISs (AFP; Supplementary Table 1). Asterisks indicate gene products that were identified by MS in purified CISs. **b**, fLM of *Anabaena* expressing Cis11–sfGFP revealed numerous foci in every cell, mainly located in the cell periphery. Autofluorescence originating from chlorophyll is visible in magenta. Shown is a central image of a confocal stack. The experiment was repeated four times with similar results and in total 29 confocal z-stacks were acquired. Bar, 10 μm. **c,e**, Slices through cryo-electron tomograms of cryoFIB-milled *Anabaena* cells showed CISs (brown arrowhead) anchored in outermost thylakoid membrane stack. CISs were found pointing to the septum (observed in 80 tomograms from 14 independent datasets), which connects adjacent sister cells in a filament (**c**), as well as pointing to the peripheral cell envelope (**e**; observed in 16 tomograms from 9 independent datasets). CM, cytoplasmic membrane; PB, phycobilisomes; PG, peptidoglycan; S, septum; TM, thylakoid membrane stack. Shown are 13.4 nm thick slices. Bar, 100 nm. **d,f**, Volume segmentations of full tomograms shown in **c** and **e** are visualizing the 3D organization, including thylakoid membranes (green), cytoplasmic membrane (dark brown), peptidoglycan (grey), storage granules (orange), cyanophycin granule (purple) and CISs (light brown). **g**, Examples and quantification of CISs (brown arrowheads) found in different cellular locations. A total of 99 tomograms from 14 independent datasets were analysed. OM, outer membrane. Shown are 13.4 nm thick slices. Bars, 100 nm. **h**, Subtomogram average of CIS baseplate region showed six wide connectors (red numbers) and six narrow connectors (yellow numbers), facilitating interactions with the TM. The 204 particles from 99 tomograms acquired in 14 independent datasets were initially selected for averaging. Bars, 10 nm. **i**, Isosurface representation of subtomogram average shown in **h**. Connectors spanned from CIS baseplate (light brown) to thylakoid membrane pore (green).

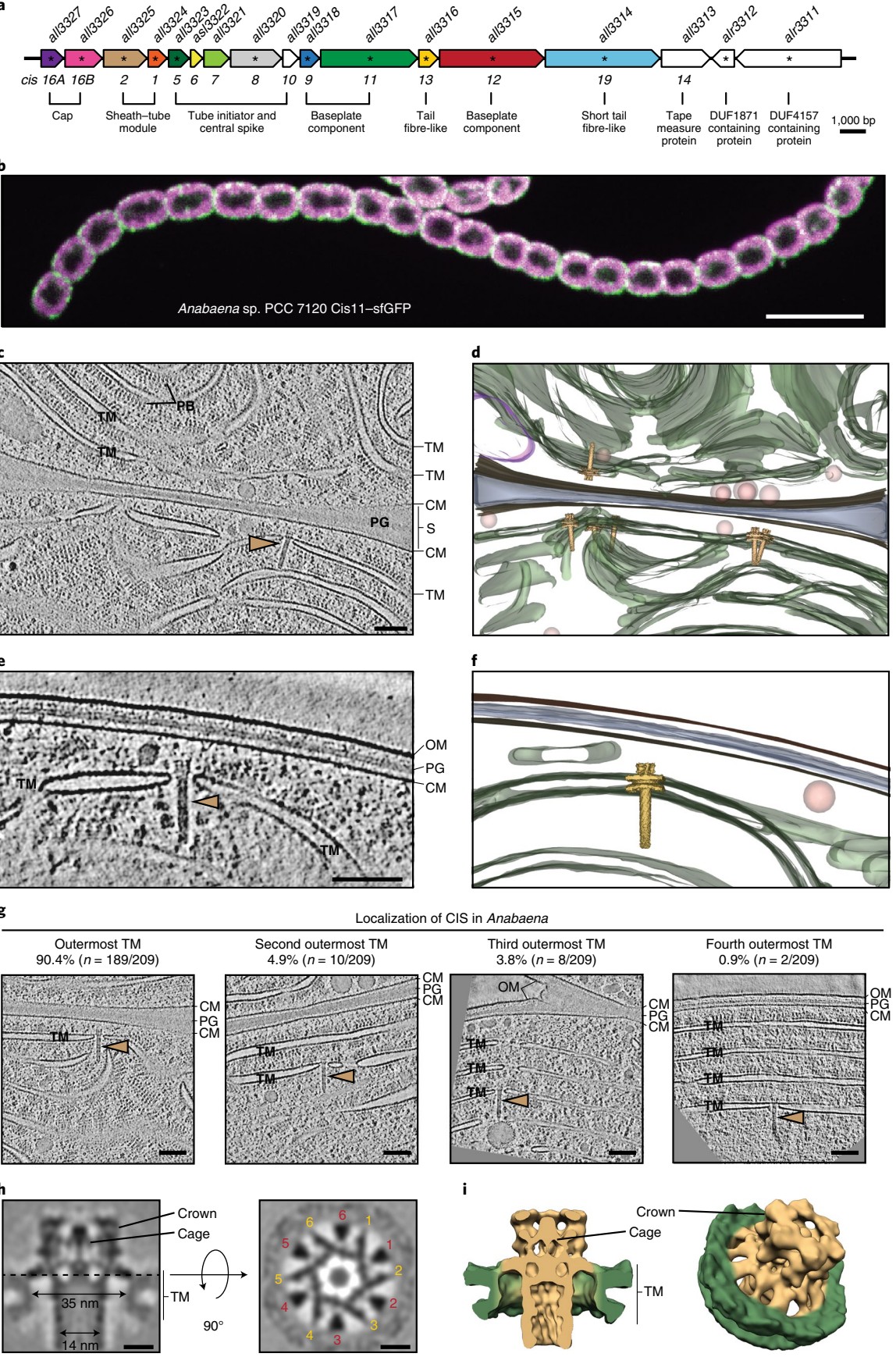

**b** *Anabaena* sp. PCC 7120 Cis11–sfGFP

**g** Localization of CIS in *Anabaena*

Outermost TM
90.4% (*n* = 189/209)

Second outermost TM
4.9% (*n* = 10/209)

Third outermost TM
3.8% (*n* = 8/209)

Fourth outermost TM
0.9% (*n* = 2/209)

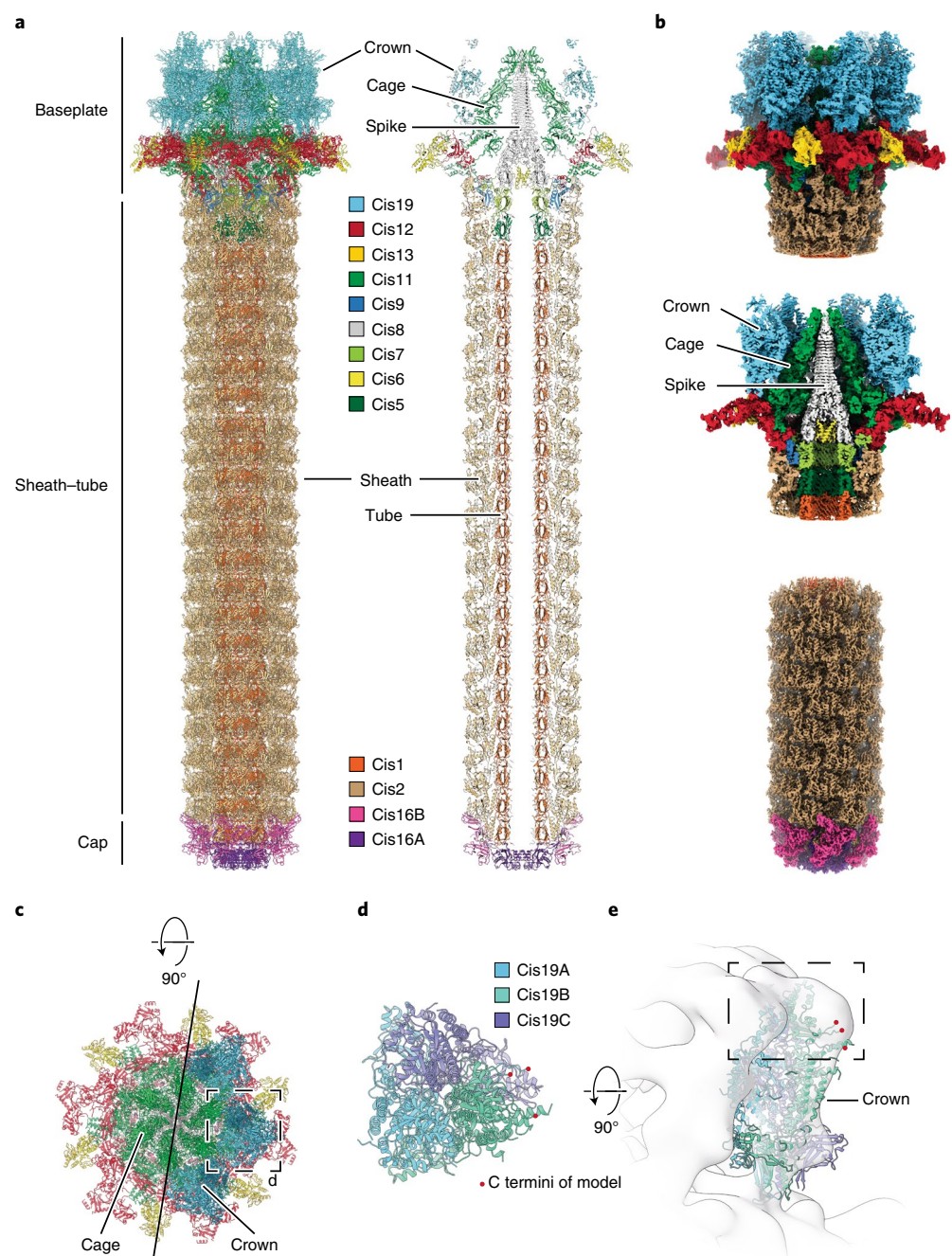

**Fig. 2 | Atomic model of CIS reveals unique features. a**, Atomic model of full CIS in side view (left) and sliced (right). The protein colour code matches the gene cluster in Fig. 1a. **b**, Surface renderings of a composite cryoEM map of baseplate (upper panel; sliced view in middle panel) and sheath/tube/cap complex (lower panel) that were reconstructed separately. Subunits have been coloured according to the atomic model in **a**. **c**, Top view of baseplate complex without (left) and with (right) crown trimers composed of Cis19. Cage and crown were not observed in previous high-resolution structures of other CISs. **d**, The crown was assembled from Cis19 trimers assuming pseudo-C3 symmetry. Shown is a magnified view of the box indicated in **c**, with only one Cis19 trimer. C termini of incomplete atomic models (red dots) congregated at the peripheral side. **e**, Fitting of the atomic model of the Cis19 trimer into the in situ structure revealed the part of the density that could not be modelled due to high flexibility (dashed box).

C-terminal alpha helix could not be completely modelled due to its structural flexibility, secondary structure prediction identified a putative transmembrane helix (tmh) coinciding with the domain contacting the membrane (Fig. 3c, Supplementary Fig. 8b and Extended Data Fig. 5).

To further investigate the role of Cis12 and Cis13 in TM anchoring, we expressed both genes heterologously in *Escherichia coli* fused to sfGFP. While the strain expressing sfGFP–Cis12 showed

fluorescence in the cytoplasm (Supplementary Fig. 9a), the strain expressing sfGFP–Cis13 showed signal localization to the bacterial cell envelope (Supplementary Fig. 9b). Strikingly, a truncated version of sfGFP–Cis13 devoid of the predicted tmh showed fluorescence signal in the cytoplasm (Supplementary Fig. 9b). The localization of sfGFP–Cis13 to the membrane was further confirmed by western blotting of the membrane fraction and the soluble lysate (Supplementary Fig. 9c). We therefore proceeded to

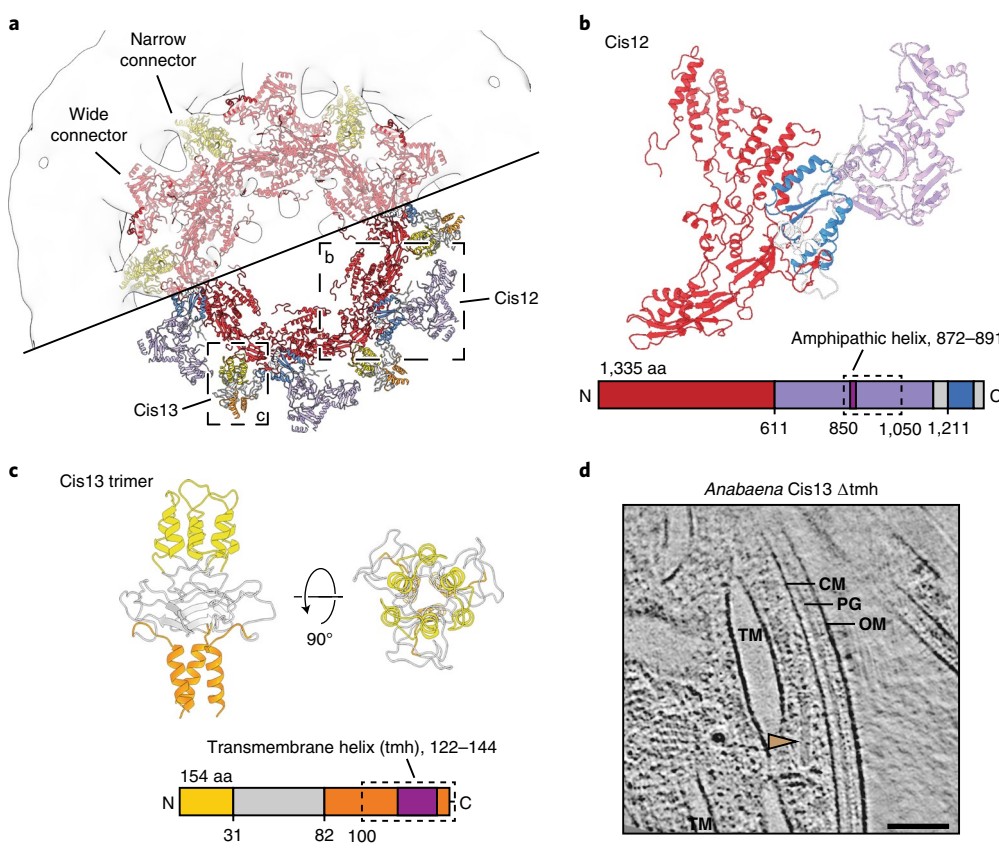

**Fig. 3 | Baseplate and tail fibre extensions mediate TM anchoring. a**, The atomic model of Cis12/Cis13 fitted into the in situ subtomogram average (top half, transparent) revealed the identity of the wide and narrow connectors to the TM. Dashed boxes indicate Cis12 and Cis13 as shown in **b** and **c**. **b**, Shown is the atomic model of Cis12 with conserved (red and blue) and extra (purple and grey) domains. Dashed box marks part of sequence that could not be modelled. Dark purple indicates sequence of a putative amphipathic helix. aa, amino acids. **c**, The atomic model of Cis13 shows the trimeric assembly of the baseplate binding N-terminal domains (yellow), an intermediary domain (grey) and the C-terminal helices (orange) extending towards the TM. The dashed box marks the part of the sequence that could not be modelled. Dark purple indicates the sequence of a putative transmembrane helix (tmh). **d**, CryoET (shown is a 13.4 nm thick slice) of cryoFIB-milled *Anabaena* cells expressing truncated Cis13 (lacking transmembrane domain) revealed free-floating CISs (brown arrowhead; 9 out of 14 CISs observed in 9 tomograms were free-floating, in total 22 tomograms were collected), which were never seen in wild-type (further examples in Extended Data Fig. 6a,b). Abbreviations as in Fig. 1. Bar, 100 nm.

generate a Cis13 mutant in *Anabaena*, deleting the transmembrane domain. In situ cryoET imaging revealed free-floating CISs (9 out of 14 CISs observed in 9 tomograms, 22 tomograms collected), which were never observed in the wild-type (Fig. 3d). The CISs that were still seen associated with the TM all appeared at unusual (non-perpendicular) angles, indicating the importance of the Cis13 transmembrane domain for the proper positioning of CISs in TM pores (Extended Data Fig. 6). Bioinformatic sequence analyses showed that the transmembrane domain of Cis13, the extension of the baseplate component Cis12 and the crown domain of Cis19 were often conserved together in other cyanobacterial CIS gene clusters (Supplementary Table 2). Interestingly, the Cis13 transmembrane domain was exclusively found in filamentous cyanobacteria, which may indicate an adaptation linked to the multicellular lifestyle.

**Stress induces ghost cell formation and exposes CISs.** Since CIS anchoring in the TM deviates from the established modes of action of eCISs and T6SSs, we further investigated under which circumstances CISs could fire and/or encounter a potential target cell. As primary producers, cyanobacteria are often the target of predatory organisms. To test a potential effect of CISs after the uptake and digestion of *Anabaena* by predators, we generated an insertion mutant strain that was deficient in CIS assembly (CIS⁻, *cis2*::pRL277). Co-incubation of wild-type and CIS⁻ *Anabaena*

cultures with potential predators, however, revealed no notable phenotypes (Supplementary Table 3). Interestingly though, in the course of these experiments, we found a subpopulation of individual, roughly spherical *Anabaena* cells, which did not show the typical red autofluorescence of chlorophyll (in figures displayed as magenta), a marker for living cells. Instead, these cells only showed an unspecific green autofluorescence. This green autofluorescence is normally superimposed by red chlorophyll fluorescence in viable cells and its observation has therefore been described as marker for dead cells[25]. We refer to these cells as 'ghost cells' since our observations of single non-viable cells were reminiscent of previous reports of programmed cell death in cyanobacteria in response to stress[26–28]. Imaging of Cis11–sfGFP-expressing *Anabaena* by fLM revealed ghost cells next to intact filaments, both exhibiting sfGFP foci, suggesting the presence of CISs in both cell types (Extended Data Fig. 7a, left, and b). We proceeded to investigate whether stress could indeed induce ghost cell formation. The treatment of fresh *Anabaena* cultures by ultraviolet (UV) light or high salt concentrations resulted in a dramatic increase in ghost cell numbers, caused by the lysis of intact cells (Fig. 4a, Extended Data Fig. 7a,c,d and Supplementary Video 2). We found that ghost cells remained stable for several weeks (Extended Data Fig. 7e).

CryoEM projection images showed the overall architecture of individual ghost cells from a UV-treated *Anabaena* culture, having

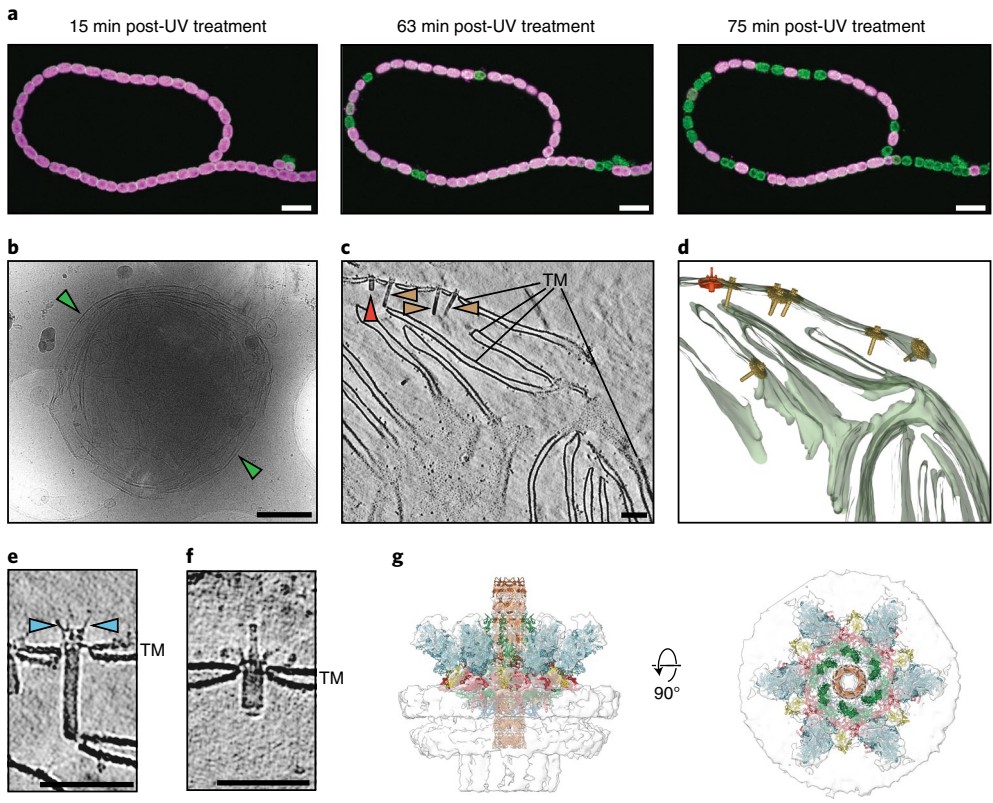

**Fig. 4 | Stress induces ghost cell formation and exposes CISs to the environment. a**, Time-lapse fLM of stressed (UV-treated) *Anabaena* cells expressing Cis11–sfGFP. Living cells showed autofluorescence (magenta) originating from chlorophyll. With increasing time, individual cells in the filament started to lyse and ghost cells (green autofluorescence) were released. For full-length time-lapse, see Extended Data Fig. 7c,d and Supplementary Video 2. The experiment was repeated four times with similar results and in total 26 time-lapse videos were acquired. Bars, 10 µm. **b**, CryoEM micrograph of an individual ghost cell. In all analysed images (>800 2D projections and 115 tomograms from 8 independent experiments), the outer membrane and the peptidoglycan cell wall were always absent. The cytoplasmic membrane was often ruptured or completely absent, exposing the TM (green arrowhead) to the environment. Bar, 1 µm. **c,d**, Slice through a cryo-tomogram of cryoFIB-milled *Anabaena* ghost cells (**c**) and the corresponding 3D volume segmentation of the full tomogram (**d**) revealed that extended CISs (brown arrowhead, 79 out of 90 CISs observed in 15 tomograms) and contracted CISs (red arrowhead; not seen in intact cells; 11 CISs observed within 6 tomograms) were directly exposed to the environment. TM, putative thylakoid membrane stack (green). Shown is a 17.2 nm thick slice in **c**. See Extended Data Fig. 8b for more ghost cell tomograms. Bar in **c**, 100 nm. **e,f**, Shown are close-up views of extended (**e**) and contracted (**f**) CISs from ghost cell cryo-tomograms. Extended CISs revealed crown protrusions (blue arrowheads; 18 CISs observed in 8 individual tomograms) that resembled small tail fibre-like structures. Contracted CISs were still anchored to the thylakoid membrane. Shown is a 17.2 nm thick slice. Bars in **e,f**, 100 nm. **g**, Atomic model of contracted CIS fitted into low-resolution in situ subtomogram average of contracted CIS in ghost cells (resolution of subtomogram average not estimated due to low particle number ($n = 11$)). The atomic model was docked by rigid-body fitting of atomic models from the extended CIS structure into the SPA map of the contracted CIS (Supplementary Figs. 5, 6 and 10). The cage domain of Cis11 (green) opened the central pore to accommodate the inner tube. The crown trimers (Cis19; blue) tilted significantly towards the baseplate periphery and showed only weak density suggesting a high degree of flexibility.

a diameter of ~3.5 µm (Fig. 4b). CryoET of cryoFIB-milled ghost cells (Extended Data Fig. 8a) revealed that ghost cells were mainly composed of TMs (Fig. 4c,d and Supplementary Video 3). In all cases ($n = 88$ ghost cells in 72 tomograms), the outer membrane and the peptidoglycan cell wall were completely absent. In about half of the ghost cells, the cytoplasmic membrane seemed to be still intact ($n = 41$ out of 88 ghost cells), whereas in the other half ($n = 47$ out of 88 ghost cells) the cytoplasmic membrane was either disrupted or completely absent. Strikingly, CISs always remained anchored in the TM and were exposed to the environment (Fig. 4c,d and Extended Data Fig. 8b).

**TM-anchored CISs present a distinct mode of action.** We then analysed the structures of CISs in ghost cells. Extended CISs often showed prominent features extending from the crown that resembled small tail fibre-like structures and were up to 15 nm in length (Fig. 4e and Extended Data Fig. 9a). Furthermore, subtomogram

averaging of ghost cell CISs (Extended Data Fig. 9b,c) showed additional densities in the crown region, compared to the subtomogram average of CISs in intact cells (Extended Data Fig. 9d,e). This could be explained by increased rigidity of these small tail fibres or enhanced contrast due to the absence of cytoplasm. Another possibility would be a conformational change in the crown region following cell lysis. These small tail fibres probably correspond to parts of Cis19 that remained unmodelled (Extended Data Fig. 9f,g) and could be involved in target binding.

Interestingly, 11 out of 90 CISs were found in a contracted state (Fig. 4f), which was never seen in intact cells. To analyse the contracted conformation and determine the fate of the crown and spike cage upon firing, we generated an in situ structure of contracted ghost cell CISs by subtomogram averaging (Fig. 4g and Supplementary Fig. 10a) and a structure of purified contracted CISs by SPA cryoEM using the existing dataset (Supplementary Fig. 10b). A final map at ~7 Å resolution for the baseplate complex

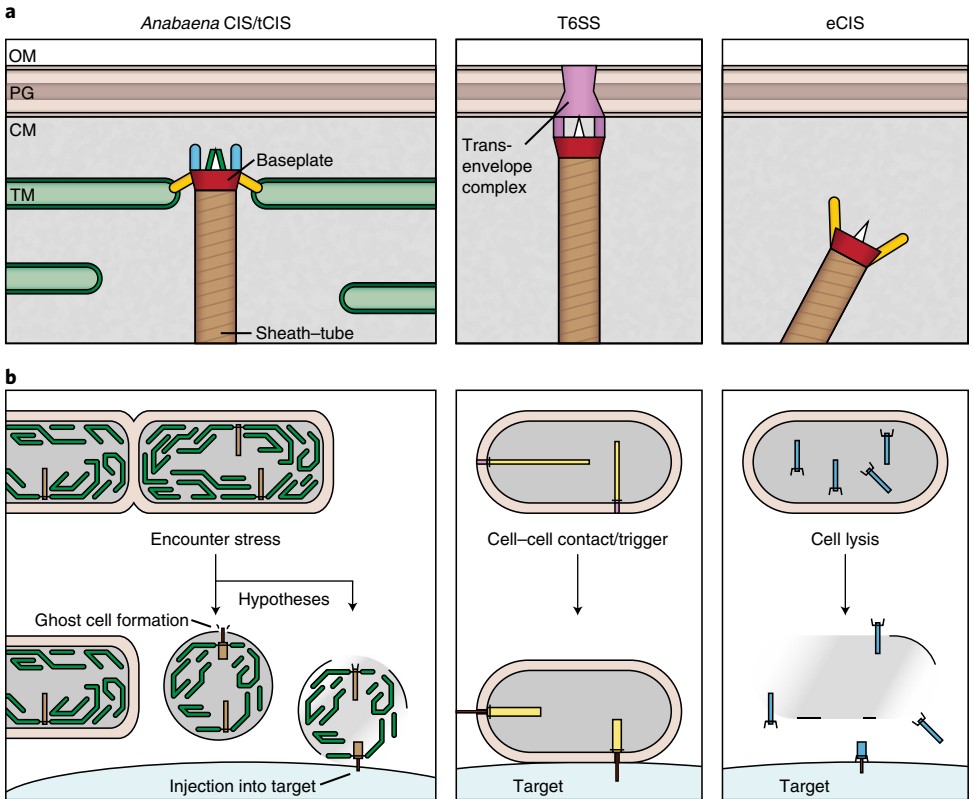

**Fig. 5 | *Anabaena* CISs (tCISs) represent a distinct mode of action.** Shown are schematics comparing *Anabaena* CISs (left), T6SSs (middle) and eCISs (right). **a,b**, Structural elements are the basis for different cellular localizations (**a**) and different modes of action (**b**). The eCISs are free-floating in the cytoplasm, released upon cell lysis and bind to their target via tail fibres (yellow). T6SSs are anchored to the CM by a trans-envelope complex and penetrate their cell envelope without compromising viability. *Anabaena* CISs, now referred to as tCISs, are anchored in the TM by a tail fibre-like protein (yellow) and an extension of a baseplate component. The tCISs act on stress and may play a role in ghost cell formation and/or mediate the interaction of ghost cells with target cells.

allowed us to dock the previously built models using rigid-body fitting. The Cis11 cage domain and the crown trimers (Cis19) tilted significantly towards the baseplate periphery, opening the central pore to accommodate the inner tube (Fig. 4g and Supplementary Fig. 10b,c). Together, these results suggest that *Anabaena* CISs can eject their inner tube by sheath contraction, however, firing only ever occurred upon cell lysis and/or ghost cell formation.

To detect a potential effect that CISs in ghost cells could have on target organisms, we co-incubated ghost cells with different bacteria and eukaryotic cells. Even though the assays did not reveal a clear difference between wild-type and CIS⁻ mutant ghost cells (Supplementary Table 4), this possibility should not be ruled out, since some CISs are known to be highly specific to certain target strains[29]. We further explored an alternative hypothesis regarding function, namely whether CISs themselves were involved in ghost cell formation. Since both wild-type and CIS⁻ mutant cells induced ghost cell formation upon stress (Supplementary Video 4), we compared ghost cell ultrastructures by cryoET imaging and by recording two-dimensional (2D) cryo-projection images. Interestingly, both experiments indicated the tendency that the CIS⁻ mutant showed a higher fraction of ghost cells with intact cytoplasmic membrane, whereas the cytoplasmic membrane was ruptured or entirely absent in most wild-type ghost cells (Extended Data Fig. 10).

Taken together, *Anabaena* CISs are closely related to other CISs (Supplementary Table 1); however, their in situ localization is inconsistent with T6SSs and eCISs (Fig. 5a). In contrast to cytoplasmic free-floating eCISs and cytoplasmic membrane-anchored T6SSs, *Anabaena* CISs are anchored to the TM, which is mediated by modified tail fibres and a baseplate component. We will refer to these TM-anchored CISs as 'thylakoidCISs' (tCISs), The unique anchoring to the TM—and more specifically their outward-pointing localization in the outermost TM—suggests a previously unknown mode of action. It is possible that the unique in situ localization has evolved to allow tCISs to function upon encountering cellular stress. We speculate (1) that tCISs may themselves play a role in ghost cell formation (for example, shedding of the cytoplasmic membrane) and/or (2) that tCISs in ghost cells are able to inject cargo into target cells, such as the many diverse organisms that are known to prey on cyanobacteria[30] (Fig. 5b).

## Conclusion

In conclusion, our integrative dataset on tCISs will serve as a framework for future studies in different fields. The atomic model pin-pointed tail fibre and baseplate components as hotspots for evolutionary re-engineering of CISs to function by different modes of action. The structure will also facilitate mechanistic understanding of closely related assemblies that function as eCIS or T6SS[iv] (refs. [10–12,15,24]). The TM-anchored in situ structure and distinct mode of action may mediate insight into cyanobacterial cell biology, in particular multicellularity and the poorly studied process of programmed cell death, which has been suggested to aid toxin release upon oxidative stress in cyanobacteria[31].

Finally, inspired by the high abundance of related CIS gene clusters in cyanobacterial genomes and the fact that cyanobacteria are

among the most abundant organisms on the planet, we wondered whether we could purify CISs directly from the natural environment. Our successful detection of cyanobacterial CISs in a water sample from Lake Zurich by MS and negative-stain EM (Supplementary Fig. 11) therefore not only suggests that studying CISs directly in environmental samples is within reach but it also further demonstrates the biological significance of CISs in the environment.

## Methods

**Bacterial strains.** *Anabaena* sp. PCC 7120 and derivates were cultivated in liquid BG11 medium at 28 °C with constant illumination at ~10 µE m⁻² s⁻¹, shaking at 120 r.p.m. or were grown on BG11 medium solidified with 1.5% (w/v) Difco agar[32]. Indicated mutant strains were cultured in BG11 media supplemented with antibiotics at the following concentrations: 50 µg ml⁻¹ neomycin, 20 µg ml⁻¹ spectinomycin and 20 µg ml⁻¹ streptomycin (Supplementary Table 5).

**Generation of *Anabaena* mutant strains.** Triparental mating (conjugation)[33] with the *E. coli* strains J53 (RP-4) (ref. [34]) and HB101 (pRL528) (refs. [34,35]) carrying the respective cargo plasmid was performed to introduce plasmids into *Anabaena* strains. Reference sequences were obtained from KEGG (https://www.kegg.jp/) and amplified via colony polymerase chain reaction (PCR) using *Anabaena* sp. PCC 7120 culture as template. Plasmid DNA and PCR fragments were purified using the Monarch Kits (New England Biolabs). Plasmids were electroporated into *E. coli* NEB 10-β (New England Biolabs) cells (1.8 kV, 25 mF, 200 Ω) and sequences were verified by sequencing (GATC biotech AG). The C-terminal sfGFP fusion to *cis11* was achieved by cloning *all3317* downstream of the putative promoter region of the *all3327–all3315* operon (400 base pair (bp) region upstream of *all3327*) translationally fused to a 5×GS-linker and sfGFP into the self-replicating plasmid pIM612 (ref. [36]). The region upstream of *all3327* was amplified using oligonucleotides 1,718/1,719, *all3317* (void of the stop codon) with oligonucleotides 1,720/1,721 and 5×GS–sfGFP with oligonucleotides 1,289/1,444 using pIM608 as template[36]. The fragments were assembled via Gibson assembly cloning[37] into EcoRI-digested pIM612 yielding plasmid pIM713. The CIS deletion strain (CIS⁻) was created by single homologous recombinant insertion of pRL277 (ref. [38]) into the coding region of *all3325* (*cis3*::pRL277). For this, an internal 864 bp fragment of *all3325* was amplified with oligonucleotides 1,722/1,723 and inserted into XhoI/PstI-digested pRL277 resulting in plasmid pIM714. Insertion into the genome was checked with oligonucleotides 1,717 (upstream of insertion)/890 (in plasmid), segregation was checked using oligonucleotides 1,717/1,716 (upstream and downstream of insertion). The mutant DR815 expressing a truncated All3316 protein (Met1-His100) was created by double-crossover homologous recombination of plasmid pIM815 into the genomic *all3316*-site of *Anabaena* sp. For this, 830 bp fragments upstream (including a stop codon) and downstream of the truncated *all3316*-region were amplified using oligonucleotides 2,311/2,312 and 2,313/2,314, respectively. Furthermore, the C.K3 kanamycin/neomycin resistance cassette[39] was amplified with oligonucleotides 1,864/2,319. The amplified C.K3 gene was lacking a promoter and terminator region to enable transcription of the genes downstream of *all3316* in the CIS operon. The three PCR fragments were ligated into a XhoI/PstI-digested pRL277 vector[38] via Gibson assembly cloning creating pIM815 (upstream fragment, C.K3, downstream fragment). Segregation of the truncated *all3316* gene was checked using oligonucleotides 2,315/2,316 creating mutant DR815.

**Purification of CISs for negative-stain EM and MS.** A total 50 ml of a well-grown *Anabaena* sp. PCC 7120 culture were pelleted, resuspended in 5 ml of lysis buffer (150 mM NaCl, 50 mM Tris-HCl, 0.5× CellLytic B (Sigma-Aldrich), 1% Triton X-100, 200 µg ml⁻¹ lysozyme, 50 µg ml⁻¹ DNAse I, 1 mM phenylmethylsulfonyl fluoride, pH 7.4) and incubated for 30 min at 37 °C. Cell debris were removed by centrifugation (15,000 *g* for 15 min at 4 °C) and cleared lysates were subjected to ultracentrifugation (150,000 *g*, 1 h, 4 °C, 2 ml of 40% sucrose cushion). Pellets were resuspended in 100 µl of resuspension buffer (150 mM NaCl, 50 mM Tris-HCl, supplemented with protease inhibitor cocktail (Roche), pH 7.4). Proteins in the CIS preparation were identified by MS at the Functional Genomics Center Zurich. A total 40 µl of CIS preparation was digested with 5 µl of trypsin (100 ng µl⁻¹ in 10 mM HCl) and microwaved for 30 min at 60 °C. The sample was dried and dissolved in 50 µl of 0.1% formic acid, diluted in 1:10 and transferred to autosampler vials for liquid chromatography with tandem mass spectrometry analysis. A total of 1 µl was injected. Database searches were performed by using the Mascot swissprot and tremble_fungi search program. For search results, stringent settings have been applied in Scaffold (1% protein false discovery rate, a minimum of two peptides per protein, 0.1% peptide false discovery rate).

**Precipitation of culture supernatant for western blotting.** Cell pellet was treated as described for negative-stain EM and MS above. Culture supernatant was precipitated with 20% trichloroacetic acid at 4 °C overnight and subsequently spun down at 18,000 *g* for 20 min. Precipitate was washed twice with 1 ml of ice-cold acetone and spun down at ~21,000 *g* for 10 min. SDS–polyacrylamide gel electrophoresis (SDS–PAGE) and western blotting were done as described below.

**Negative-stain EM.** Samples were adsorbed to glow-discharged, carbon-coated grids for 60 s, washed twice with water and stained with 2% phosphotungstic acid for 45 s. The grids were examined using a Thermo Fisher Scientific Morgagni transmission electron microscope operated at 80 kV.

**Light microscopy.** For sample preparation, 10 µl of untreated or treated *Anabaena* cultures were spotted on a BG11 agar plate and left until the liquid was fully absorbed. An area was cut out and mounted on a cell culture imaging dish. The z-stacks were collected using a ×100 objective on a Nikon Eclipse T1 microscope, equipped with a spinning disk module (Visitron) and two EMCCD Andor iXon Ultra cameras (1,024 × 1,024 pixels², 13 × 13 µm² pixel size). Images were captured using the Visiview software (Visitron) and analysed using Fiji[40]. GFP-foci in *Anabaena* Cis11–sfGFP were quantified in 129 individual cells using FIJI's foci counter. For time-lapse imaging of *Anabaena* filaments after UV treatment, one image was recorded every 2 min over 1 h using hardware autofocus. *E. coli* expressing *sfgfp–cis12* constructs were imaged as described above, whereas *E. coli* expressing *sfgfp–cis13* constructs were imaged on a thin BG11 1% agarose pad on a Zeiss LSM 880 Airyscan confocal light microscope. Time-lapse imaging comparing *Anabaena* wild-type and CIS⁻ mutant strains were recorded using a ×40 objective on a Leica Thunder Imager 3D Cell Culture equipped with a Leica DFC9000 GTC CMOS camera (2,048 × 2,048 pixels², pixel size 6.5 µm). Images were recorded every 5 min over a time course of 3 h. Images for ghost cell comparison of wild-type and Cis11–sfGFP were recorded on the same Leica Thunder Imager using a ×100 objective with additional computational clearing (Thunder) using the LasX software.

**Plunge freezing for cryoET.** Cyanobacterial cultures sedimented for ~45 min and were concentrated by removing two-thirds of the medium. Ghost cells were prepared as described below. A total 3.5 µl of cell suspensions were applied to glow-discharged copper or molybdenum EM grids (R2/2, Quantifoil), automatically blotted and plunged into liquid ethane/propane[41] using a Vitrobot Mark IV (Thermo Fisher Scientific)[42]. Using a Teflon sheet on one side, all samples were blotted exclusively from the back for 4–6 s (ref. [43]). Frozen grids were stored in liquid nitrogen until loaded into the microscope.

**CryoFIB milling.** CryoFIB milling[19,21] was done according to Medeiros et al.[44] and Weiss et al.[20]. Plunge-frozen grids were clipped into cryoFIB-autoloader grids (provided by J. Plitzko, Max Planck Institute of Biochemistry) and mounted into a 40° pretilted scanning electron microscopy (SEM) holder (Leica Microsystems). The holder was transferred with a VCT100 cryo-transfer system (Leica Microsystems) and inserted into a custom-built cryo-stage within a Helios NanoLab600i dual beam FIB/SEM microscope (Thermo Fisher Scientific) controlled by the Thermo Fisher Scientific XT software. Grids were coated with platinum precursor gas for 6 s. SEM at 3–5 kV (80 pA) was used to identify targets and 8–10 µm wide lamellae were prepared in multiple steps using the focused gallium ion beam. The ion beam current was gradually reduced from 43 nA to 24 pA according to lamella thickness. After a final lamella thickness of ~250 nm was achieved, the holder was returned to the loading station with the VCT100 transfer system. Unloaded grids were stored in liquid nitrogen.

**CryoET.** CryoFIB-thinned cyanobacteria and UV-induced ghost cells were imaged by cryoET. Micrographs were recorded on Titan Krios 300 kV FEG transmission electron microscopes (Thermo Fisher Scientific) equipped with Quantum LS imaging filters (slit width 20 eV) and K2 or K3 direct electron detectors (Gatan). A low magnification overview of the grid was recorded using SerialEM[45,46]. Tilt series were collected automatically using UCSF Tomo[47] or SerialEM and covered an angular range from −50° to +70° with 2° increment for cryoFIB-milled lamellae with a defocus of −6 to −9 µm. The total dose of a tilt series accumulated 140–160 e⁻ Å⁻² and the pixel size at the specimen level was 3.38 Å for cyanobacterial cells or 4.3 Å for ghost cells. The 2D projection images of ghost cells were recorded at a pixel size of 16.77 or 18.85 Å.

**Tomogram reconstruction and subtomogram averaging.** Tilt series were drift-corrected using alignframes and CTF correction and three-dimensional (3D) reconstructions were generated using the IMOD package[48,49]. For visualization, tomograms were filtered using the tom_deconv deconvolution filter[50]. CIS structures were manually identified in individual tomograms and their long axes from baseplate to cap were modelled with open contours in 3dmod[49] to generate model points, the initial motive list and particle rotation axes[43]. Initial averages that were used as subsequent first references were created with PEET[51]. Particle extraction, alignment and averaging were done with Dynamo[52]. The 4 × 4-binned particles were aligned for eight iterations, limiting resolution to 50 Å (box size 40 × 40 × 40 pixels³). Afterwards the datasets were split in half for gold-standard Fourier shell correlation (FSC) calculations and independent alignments were repeated for eight iterations using 2 × 2-binned tomograms (box sizes 80 × 80 × 80 pixels³ for CISs within intact cells; 62 × 62 × 62 pixels³ for CISs in ghost cells or contracted CISs). Following eight alignment iterations of unbinned data, the set of particles was cleaned according to cross-correlation values and another eight alignment iterations were performed. A total of 204 CISs inside intact

cells were initially selected from 99 tomograms. The final, sixfold symmetrized average, resulted from 170 particles with a box size of $160 \times 160 \times 160$ pixels[3] at a pixel size of 3.38 Å. The final, sixfold symmetrized average of CISs in ghost cells was determined from 64 particles (78 particles initially picked from 15 tomograms) with a box size of $124 \times 124 \times 124$ pixels[3] at a pixel size of 4.3 Å. Gold-standard FSC curves for resolution estimation were calculated with Dynamo after alignment of the half-maps in UCSF Chimera[53] using the fitmap function. To compare CISs within intact cells and within ghost cells, both averages were re-aligned with a mask focusing on the crown region, low-pass filtered to 40 Å and a difference map was created using the DIFFMAP software package from the Grigorieff laboratory (https://grigorieflab.umassmed.edu/diffmap). The subtomogram average of contracted CISs was calculated from 11 particles selected from six tomograms. The final average used a box size of $62 \times 62 \times 62$ pixels[3] with a pixel size of 8.6 Å. Due to the low particle number, no FSC curve was determined. The 3D rendering, segmentations and videos were done with IMOD or UCSF Chimera. Subtomogram averages (EMD-13773–EMD-13775) as well as example tomograms (EMD-13770–EMD-13772) have been deposited in the Electron Microscopy Data Bank.

**Purification of CIS particles for cryoEM.** *Anabaena* sp. PCC 7120 was cultivated in 1 l of culture for ~3 weeks. Cells were pelleted at 7,000g resulting in about 2 g of cells. After resuspension in lysis buffer (50 mM Tris-HCl pH 7.5, 150 mM NaCl), protease inhibitor cocktail (cOmplete, Roche) and DNAse I were added. Cells were lysed by sonication ($15 \times 30$ s) on ice and cell debris was spun down for 10 min at 10,000g. Membranes containing CISs were pelleted for 1 h at 90,000g, resuspended in lysis buffer and homogenized using a Dounce homogenizer. To solubilize CIS particles, *n*-dodecyl-β-D-maltopyranoside (DDM) was added to 1% and alpha-amylase was added to digest glycogen storage granules. After incubation for 15 min at room temperature the sample was spun for 1 h at 150,000g through 1 ml of 50% sucrose cushion. The pellet was resuspended in 100 µl of lysis buffer with 1% DDM and loaded on a 10–50% continuous sucrose gradient. After spinning for 12 h at 50,000g, fractions containing CISs were pooled and pelleted for 1 h at 150,000g. The pellet was resuspended in ~12 µl of lysis buffer with 0.05% DDM and used for cryoEM experiments.

**Single particle cryoEM.** CryoEM samples were prepared by plunge freezing in a Vitrobot Mark IV (Thermo Fisher Scientific). A total 3 µl of sample was applied to Quantifoil R2/2 Cu 200 mesh (Quantifoil) grids coated with a continuous layer of amorphous carbon. Grids were blotted between 8 and 10 s and plunged into liquid ethane–propane mix (37% (v/v) ethane). Single particle data were collected on four grids of two independent samples (dataset 1 and dataset 2) using a Titan Krios (Thermo Fisher Scientific) transmission electron microscope equipped with a Quantum LS imaging filter (Gatan, slit with 20 eV) and K2 direct electron detector (Gatan). Data were collected using SerialEM 3.7 (refs. [45,46]) using beam-image shift to collect four shots per hole. Pixel size at specimen level was 1.1 Å and target defocus ranged from 0.9 to 1.5 µm. Total dose per acquisition was about $52\,e^- \text{Å}^{-2}$ and was fractionated in 50 frames (Supplementary Table 6).

Frames were aligned and dose-weighed using alignframes of IMOD[54] and CTF was estimated using gctf[55]. Manual picking of a subset of extended CIS baseplates and subsequent 2D classification in RELION 3.0 (ref. [56]) resulted in 2D classes that were used for auto-picking of all micrographs.

Further rounds of 2D classification, 3D autorefine and CTF refine resulted in a map of the baseplate complex with applied C6 symmetry at a resolution of 2.9 Å (FSC = 0.143).

C3 symmetry relaxation[57] followed by 3D classification and 3D autorefine focused on the spike complex yielded a map at a resolution of 3.2 Å. To better resolve crown and baseplate periphery the symmetry was relaxed to C1. The 3D classification and 3D autorefine focused on crown and baseplate periphery yielded masked maps at resolutions of 2.9 and 3.0 Å, respectively.

Capped ends of CISs were autopicked using previous 2D classes as template. We used 2D classification, 3D classification, 3D autorefinement and CTF refinement and a map at a resolution of 2.8 Å was obtained (Supplementary Figs. 4 and 5).

Contracted particles were picked from the same dataset using crYOLO[58] trained on about 600 manually picked particles. The 2D classification, initial model generation, 3D classification, 3D autorefine and CTF refine using RELION 3.0 yielded a map at a resolution of 6.7 Å masked for the baseplate complex.

**Atomic model building.** Most proteins were built de novo using COOT[59]. Different maps were fitted to the C6 map of the baseplate complex to ensure model coherence and models were built using all maps. Cap complex proteins Cis16A and Cis16B as well as the sheath–tube module composed of Cis1 and Cis2 were built in the cap complex map. Models were iteratively refined using RosettaCM[60] and real-space refinement implemented in PHENIX[61]. Several proteins could only be partially modelled and in some cases side chains were not assigned.

Final model validation was done using MolProbity[62] implemented in PHENIX and results were presented in Supplementary Tables 7 and 8. Correlations of models and maps shown in Supplementary Fig. 5 and Supplementary Table 8 were calculated using mtriage[63].

Rigid domains of proteins modelled in the extended structure could be fitted into the low-resolution contracted map using real-space refinement in PHENIX. All visualizations were done using UCSF Chimera[53] or ChimeraX[64].

**Expression of cyanobacterial proteins in *E. coli*.** Constructs of pET15b vectors for the expression of 6×His-tagged Cis12 and Cis13 were generated by GenScript. Generation of sfGFP insertions and domain deletions (truncated sfGFP–Cis13 (Δtmh; only comprising Met1-His100 of All3316)) was done using standard molecular biology techniques[35,37]. Primers listed in Supplementary Table 9 were purchased from Microsynth AG. Plasmid manipulations were verified by DNA sequencing conducted by Microsynth AG. *E. coli* BL21Star strains were grown in lysogeny broth (LB) media. Plasmid-containing strains were grown with ampicillin at 100 µg ml⁻¹.

For expression, *E. coli* BL21Star cells were transformed with pET15b plasmids containing the desired gene construct. Cells were cultured for 1–2 h in 20 ml of LB at 37 °C until they reached an optical density $OD_{600}$ of ~0.4–0.6. Cultures were induced with 1 mM isopropyl β-D-1-thiogalactopyranoside and incubated for 2 h at 30 °C. Cells were either used for light microscopy or harvested by pelleting. Cell pellets were resuspended in lysis buffer (50 mM Tris-HCl pH 7.5, 150 mM NaCl) and lysed by sonication. Cell debris was removed by centrifuging for 15 min at 12,000g. Membranes were pelleted for 200 min at 30,000g. The membrane pellet was resuspended in Laemmli sample buffer (Bio-Rad). Membranes and supernatant were run on an SDS–PAGE gel (Bio-Rad) and western blotted using horseradish peroxidase (HRP)-conjugated anti-6×His (no. MA1-21315-HRP, Invitrogen).

**SDS–PAGE and western blotting.** SDS–PAGE was conducted using the standard protocol[65]. The samples were denatured for 5 min at 95 °C in 1× Laemmli sample buffer (Bio-Rad) before loading on a 4–15% gradient precast protein gel (Bio-Rad). The electrophoresis was performed at a constant voltage of 200 V in SDS running buffer (Tris/Glycine/SDS, Bio-Rad) for about 30 min.

The gels used for western blotting were transferred onto nitrocellulose membrane. Membranes were blocked with 5% milk in TBS-T (50 mM Tris-HCl pH 7.6, 150 mM NaCl, 0.1% Tween-20) at 4 °C for 1 h or overnight. For His-tagged samples, membranes were incubated with 1:5,000 HRP-conjugated anti-6×His (no. MA1-21315-HRP, Invitrogen) in 1% milk in TBS-T for 1 h. Alternatively, membranes were incubated with 1:1,000 polyclonal rabbit anti-all3324 (anti-Cis1) or anti-all3325 (anti-Cis2) (GenScript) and 1:5,000 secondary HRP-conjugated goat anti-rabbit IgG (Abcam) in 1% milk in TBS-T for 1 h, respectively. Membranes were washed three times for 10 min in TBS-T between and after antibody incubations. Signals were detected using a chemiluminescent substrate (ECL, Amersham).

**Ghost cell induction with UV light and salt treatment.** For the induction of ghost cell release, 10 ml of *Anabaena* cultures were transferred into a Petri dish and treated with two Sankyo Denki Germicidal 68 T5 UV-C lamps for 6–8 min in a repurposed Herolab UV DNA Crosslinker CL-1. To analyse the stability of ghost cells, the treated cell culture was incubated at room temperature over 2 weeks. Regularly, an aliquot of the culture was mixed with 5 µg ml⁻¹ of FM1-43 membrane dye and analysed by light microscopy. To stimulate ghost cell release with high salt concentration, 0.1 M NaCl was added to *Anabaena* cultures and incubated at 28 °C with constant light illumination.

**Structure prediction and sequence conservation of distinct features.** Sequences of Cis12 (All3315) and Cis13 (All3316) were analysed for putative amphipathic helixes and transmembrane domains using online tools like HELIQUEST[66] or the TMHMM Server v.2.0, respectively[67].

Sequences of the Cis12 extension not present in AFP (amino acids 610–1,100), the Cis13 predicted transmembrane domain (amino acids 122–154) and Cis19 were used for BLASTP searches against all gene clusters of the database of eCISs (http://www.mgc.ac.cn/dbeCIS/; ref. [16]) and some additional strains (marked with * in Supplementary Table 2).

**Water samples from Lake Zurich.** Water samples from Lake Zurich were collected at the centre of the lake in front of Thalwil, using a twin plankton net ($2 \times 100$ µm² mesh) tow from a depth of 10 m to the surface. A total 200 ml of this sample was then stored at 4 °C and used for CIS purification.

**Reporting Summary.** Further information on research design is available in the Nature Research Reporting Summary linked to this article.

## Data availability
Example tomograms (EMD-13770–EMD-13772), subtomogram averages (EMD-13773–EMD-13775) and SPA cryoEM maps (EMD-12029–EMD-12034; Supplementary Table 6) have been uploaded to the Electron Microscopy Data Bank. Atomic coordinates of the baseplate (7B5H) and cap complex (7B5I) have been uploaded to the Protein Data Bank. Source data are provided with this paper.

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

## Acknowledgements

We thank P. Tittmann, M. Peterek and C. Zaubitzer (ScopeM, ETH Zurich) for technical support as well as ScopeM for instrument access at ETH Zurich. T. Schwarz (ScopeM, ETH Zurich) is acknowledged for help with light microscopy. We thank A. Vagstad and J. Piel (ETH Zurich) for providing cyanobacterial cultures for preliminary observations. R. Danev (University of Tokyo) is acknowledged for input on SPA cryoEM data collection and A. Picenoni (ETH Zurich) for help with carbon coating of grids. We thank M. Kikkawa (University of Tokyo) for providing laboratory equipment for optimization of CIS purification and A. Jomaa (ETH Zurich) for input on purification procedure. C. Ericson (ETH Zurich) is acknowledged for help with insect cells and wax moth larvae during the conduction of killing assays, as well as D. Ebert and J. Hottinger (University of Basel) for their help with feeding *Anabaena* to *Daphnia* cultures. We thank T. Kozakiewicz and F. Pomati (EAWAG, Zurich) for providing water samples from Lake Zurich. L. Eberl, Y.-C. Chen and M. Pinto (University of Zurich) are acknowledged for fruitful discussions on stress conditions as well as for initial instrument access at the University of Zurich. M. Horn (University of Vienna) is acknowledged for initial discussions and D. Böck (University of Zurich) for help in analysing light microscopy data. G.L.W. was supported by a Boehringer Ingelheim Fonds PhD fellowship. F.E. received an exchange fellowship from the Japan Society for the Promotion of Science (JSPS, no. GR18104). M.P. was supported by the Swiss National Science Foundation (no. 31003A_179255), the European Research Council (no. 679209) and the NOMIS foundation. Work in Tübingen was supported by Deutsche Forschungsgemeinschaft (SFB766 and GRK1708).

## Author contributions

G.L.W conducted all experiments that include cryoFIB milling and cryoET. F.E. optimized CIS purification, collected and processed SPA cryoEM data and built the atomic model. J.X. supported SPA processing and provided an initial model for the sheath–tube module. A.-K.K. created cyanobacterial mutants and contributed light microscopy data. Ghost cell production and imaging were done by G.L.W. F.E. conducted the heterologous expression of cyanobacterial proteins in *E. coli*. G.L.W., H.M. and M.G. performed bioinformatic analysis of cyanobacterial genomes. M.G. and M.F. supported development of the CIS purification protocol. G.L.W, F.E., A.-K.K., I.M., K.F. and M.P. designed experiments. G.L.W., F.E. and M.P. wrote the manuscript with comments from all authors.

## Competing interests

The authors declare no competing interests.

## Additional information

**Extended data** is available for this paper at https://doi.org/10.1038/s41564-021-01055-y.

**Correspondence and requests for materials** should be addressed to Martin Pilhofer.

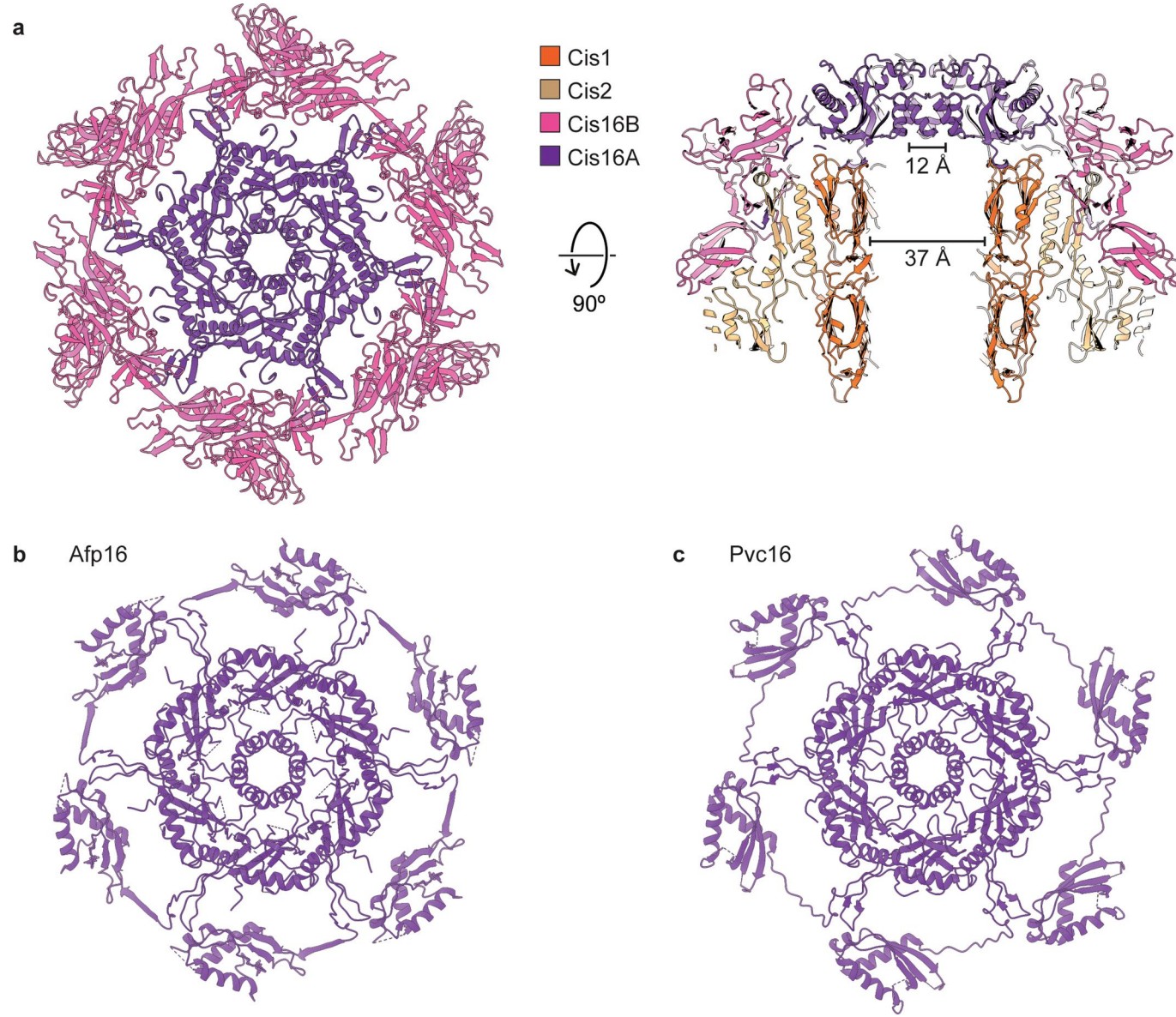

**Extended Data Fig. 1 | Distal cap complex is composed of two proteins. a:** Shown is the atomic model of the *Anabaena* CIS cap complex, composed of a hexamer of Cis16A and a hexamer of Cis16B. The complex terminates the sheath (Cis2) and tube (Cis1). **b/c:** For comparison, shown are atomic models of the cap complexes of AFP (b, PDB: 6RAP[10]) and PVC (c, PDB: 6J0F[11]). These cap complexes are comprised of only one protein (Afp16 and Pvc16, respectively).

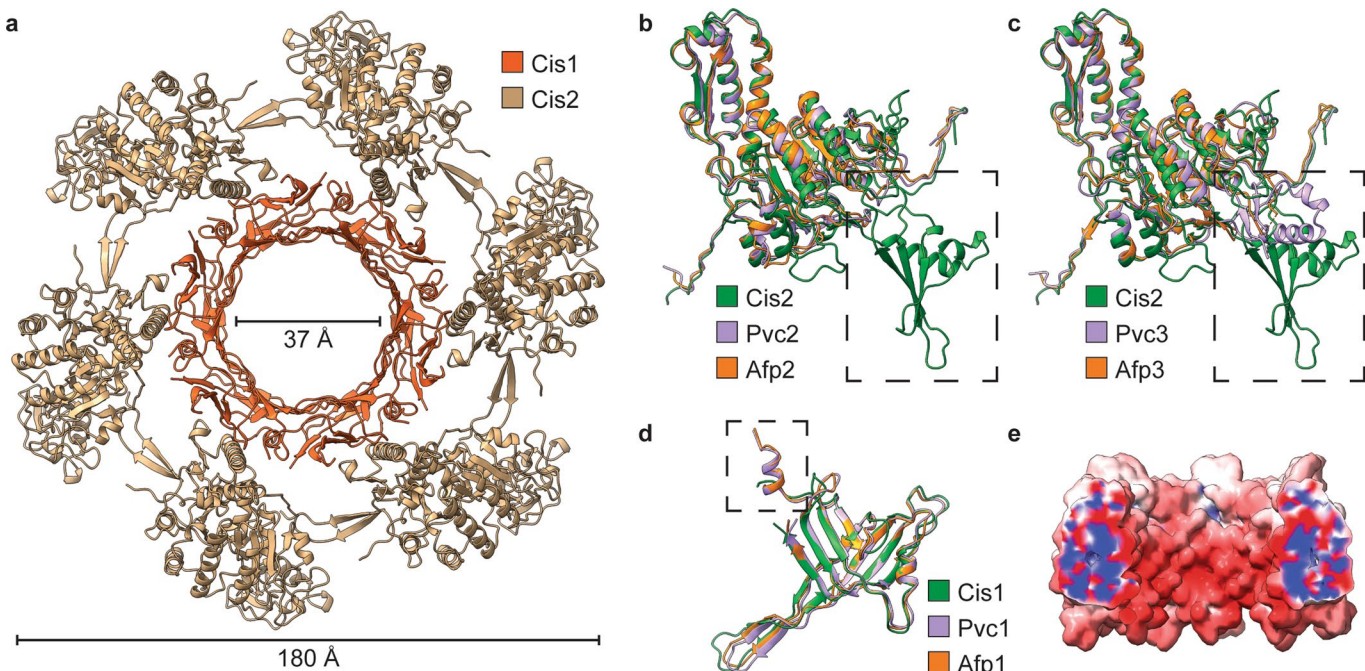

**Extended Data Fig. 2 | Sheath–tube module is highly conserved. a:** Atomic model of sheath–tube module composed of a hexamer of Cis1 surrounded by a hexamer of Cis2. **b:** The superposition of atomic models of Cis2 of *Anabaena* (green), Pvc2 of PVCs (purple, PDB: 6JOB) and Afp2 of AFPs (orange, PDB: 6RBN) exposed an extra peripheral domain in Cis2 (dashed box). **c:** The superposition of atomic models of Cis2 of *Anabaena* (green), Pvc3 of PVCs (purple, PDB: 6JON) and Afp3 of AFPs (orange, PDB: 6RBN) showed conserved positions of peripheral domains between CISs but different architecture (dashed box). **d:** The superposition of atomic models of Cis1 of *Anabaena* (green), Pvc1 of PVCs (purple, PDB: 6JOB) and Afp1 of AFPs (orange, PDB: 6RBN) showed high structural similarity. A small N-terminal helix is truncated in *Anabaena* (dashed box). **e:** Surface charge distribution of inner tube (Cis1) hexamer displaying negative (red), positive (blue) and neutral (white) charges. The surface of the tube lumen is charged negatively. Similar charge distributions have been observed in previously studied tubes of AFP and PVC.

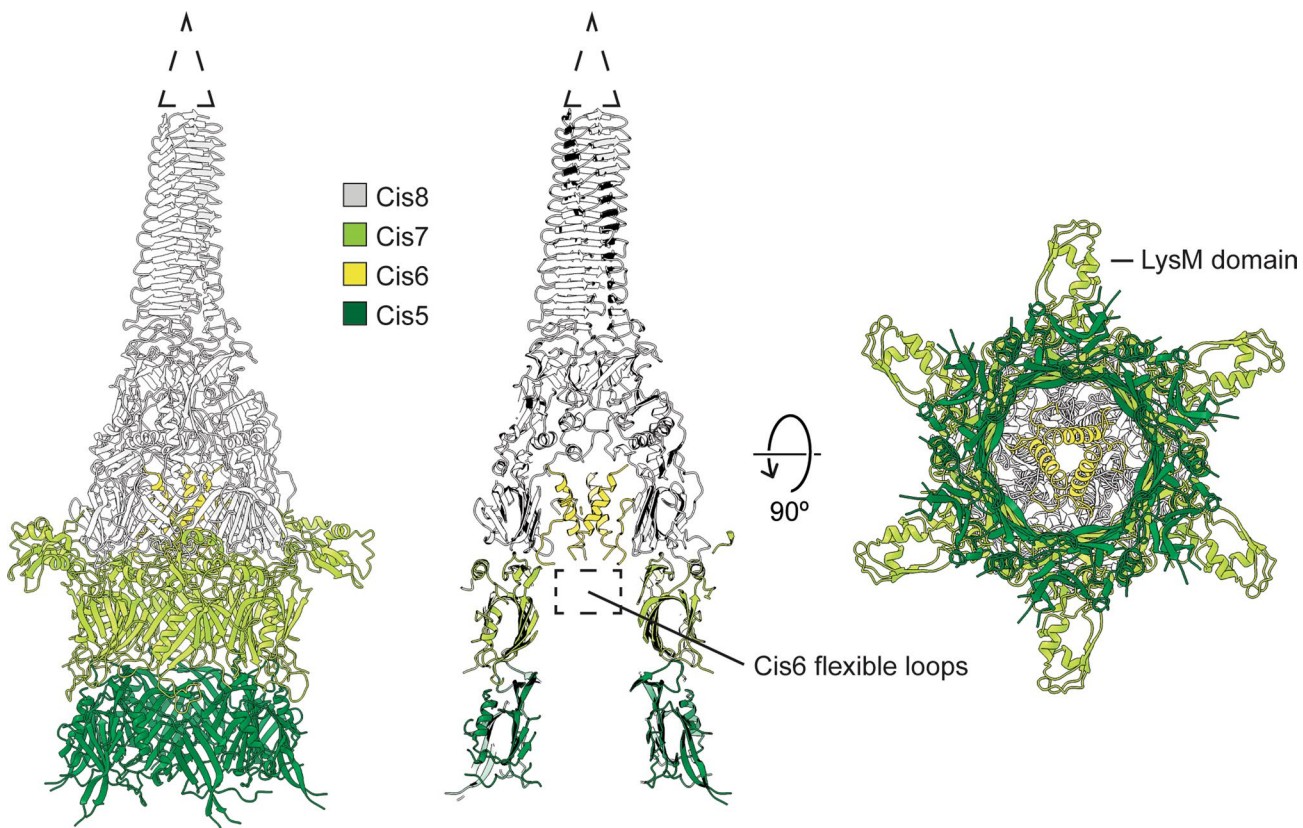

**Extended Data Fig. 3 | Model of the spike and tube initiator.** The spike complex (Cis8) and tube initiators (Cis7 and Cis5) housed a trimer of Cis6 that seals the cavity of the spike. An atomic model of Cis6 could only be built partially. A flexible loop reached into the inner tube lumen (dashed box). The spike was tipped by an additional density that could not be modelled due to a symmetry mismatch and was assigned to the PAAR-like protein Cis10 (dashed triangle). The LysM domain of Cis7 bound the inner tube to baseplate proteins Cis9 and Cis12.

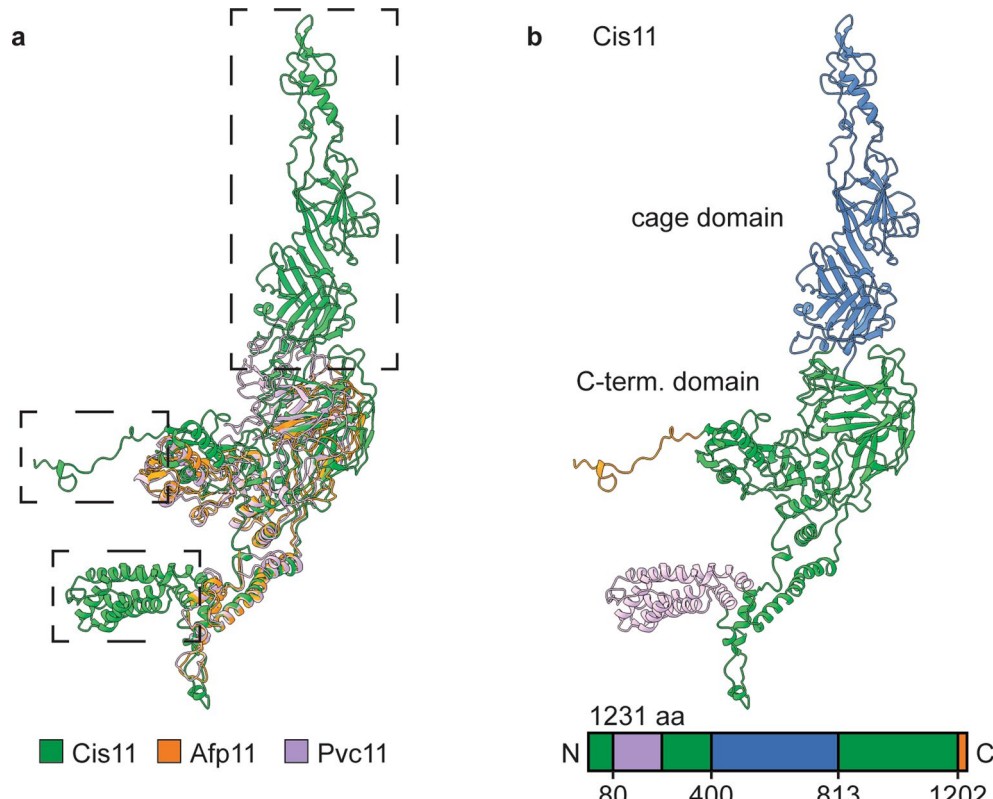

**Extended Data Fig. 4 | Cis11 contains three extra domains. a:** Shown is an atomic model of Cis11, revealing additional domains (dashed boxes) compared to Afp11 of AFP (orange, PDB: 6RAO) and Pvc11 of PVC (light purple, PDB: 6J0N). **b:** Shown is an atomic model of Cis11, highlighting the extra domains and their location within the amino acid sequence. The C-terminal domain (orange) contacted Cis13. The cage domain (blue) enveloped the spike complex. The third extra domain (light purple) offered additional interaction sites with Cis12.

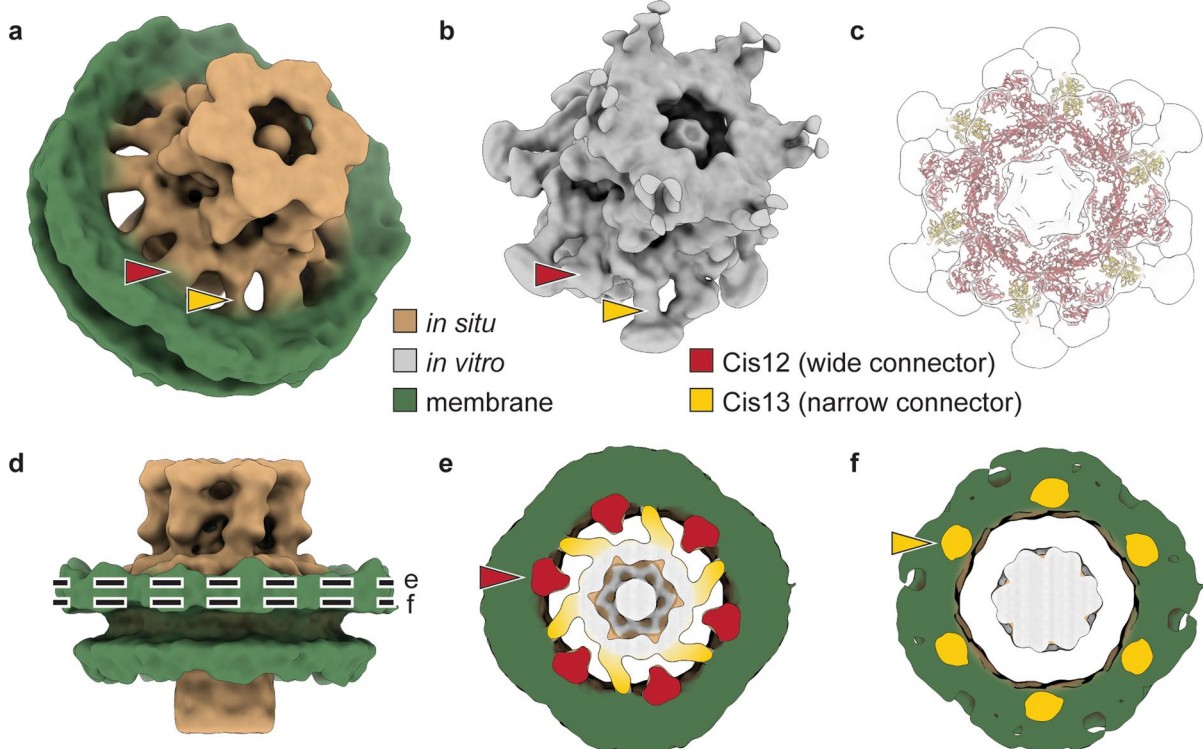

**Extended Data Fig. 5 | Cis12 contacts thylakoid membrane, while Cis13 penetrates it. a:** Isosurface of the *in situ* subtomogram average showing connectors between baseplate (brown) and thylakoid membrane (green). **b:** Low-pass filtered *in vitro* SPA map, revealing densities corresponding to Cis12 (red arrowhead) and detergent micelle surrounding transmembrane helices of Cis13 (yellow arrowhead). **c:** Overlay of atomic models of Cis12 and Cis13 and low-pass filtered *in vitro* structure illustrated peripheral parts of proteins that could not be modelled. **d-f:** Slices at different heights through *in situ* CIS architecture in *Anabaena* (indicated with dashed lines in (d) and overlayed with *in vitro* maps (e/f). *In vitro* map connectors were coloured according to protein association. Cis12 (red) contacted thylakoid membrane surface (e, red arrowhead). Cis13 (yellow) penetrated the TM and the transmembrane micelle overlapped with thylakoid membrane (f, yellow arrowhead).

*Anabaena* PCC 7120 Cis13 Δtmh

**a**

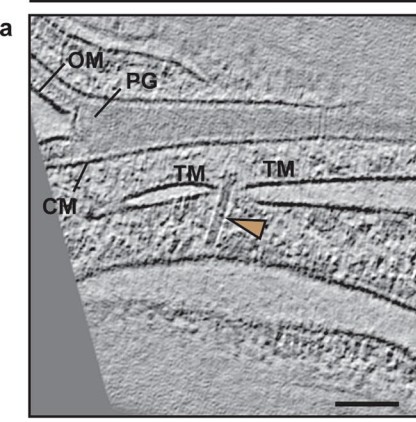 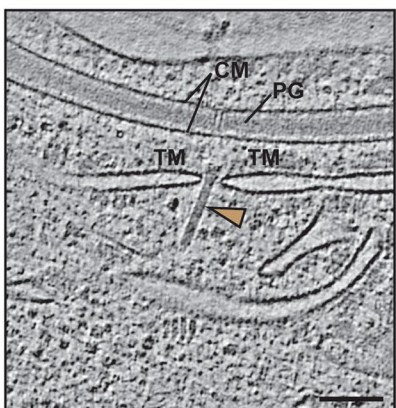 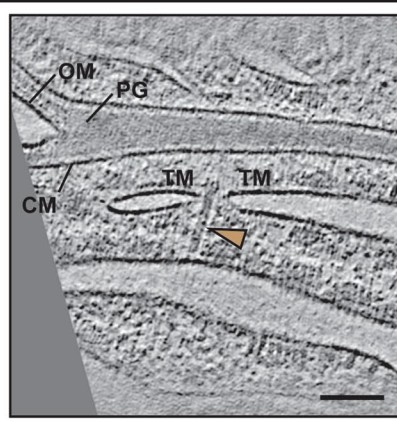

**b**

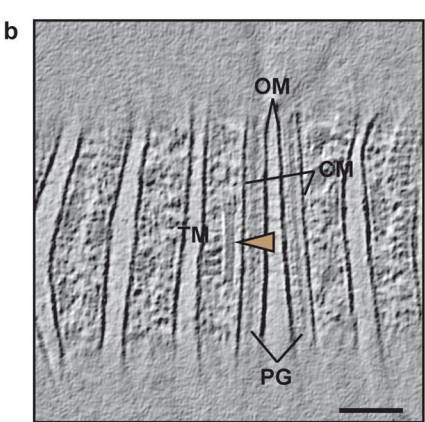 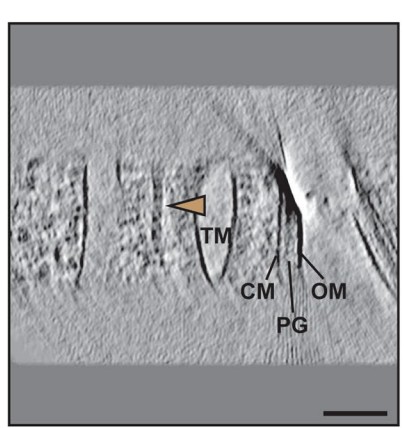 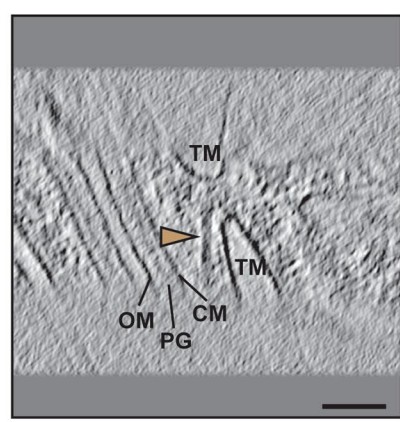

**c**

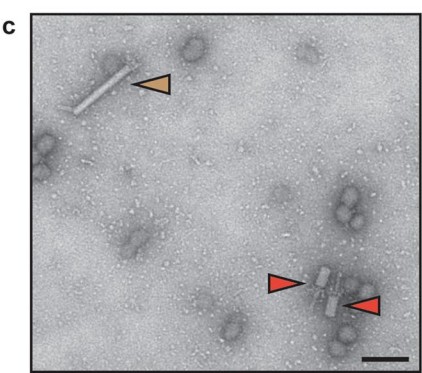 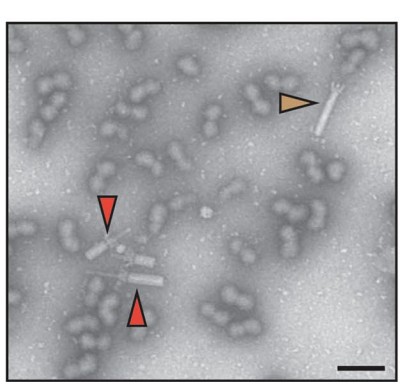

**Extended Data Fig. 6 | CISs lacking the transmembrane domain of Cis13 are free floating or loosely attached. a/b:** Slices through cryo-electron tomograms of cryoFIB-milled *Anabaena* cells expressing truncated Cis13 revealed CISs (brown arrowhead) loosely attached to thylakoid membranes (a, brown arrowheads, five CISs observed in two tomograms of individual cells) and free-floating CISs (b, brown arrowheads, nine CISs in seven tomograms of individual cells), which were never observed in wild-type *Anabaena*. CM, cytoplasmic membrane; OM, outer membrane; PG, peptidoglycan; TM, thylakoid membrane stack. Shown are a 13.4 nm thick slices. Bars, 100 nm. **c:** Negative-stain electron micrographs of CISs purified from *Anabaena* cells expressing truncated Cis13 (lacking transmembrane domain) showed CISs in extended (brown arrowheads) but also in contracted state (red arrowheads). Ten micrographs with similar results have been collected. Bars, 100 nm.

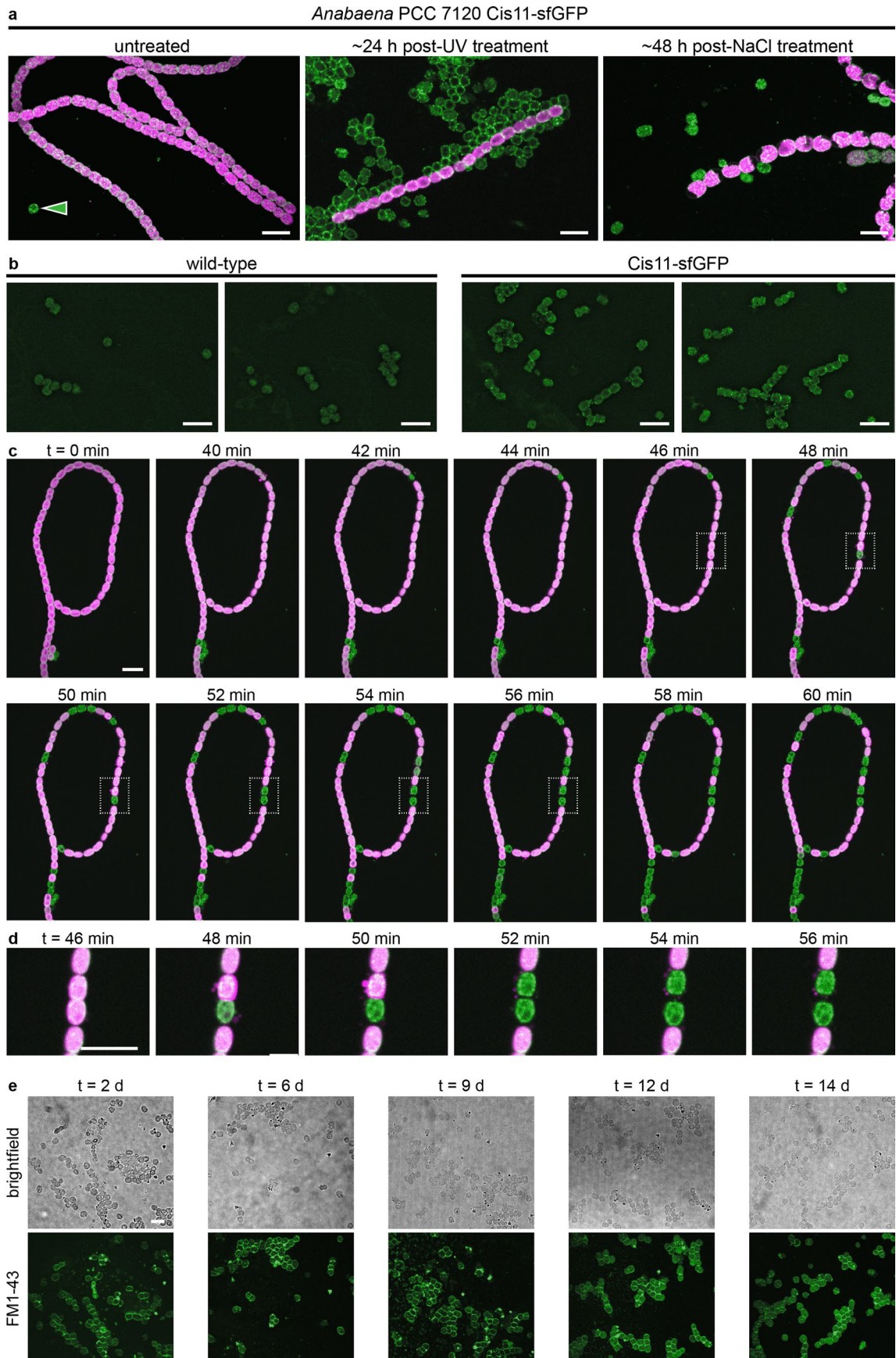

**Extended Data Fig. 7 | See next page for caption.**

**Extended Data Fig. 7 | Stress induces ghost cell formation. a:** fLM of *Anabaena* expressing Cis11–sfGFP with and without stress inducing conditions. Living cells showed red autofluorescence (excited at 575 nm, shown in magenta) originating from chlorophyll. While in untreated cultures, only few cells lost their red autofluorescence and only showed green autofluorescence [indicative of dead cells[25]] but with Cis11–sfGFP foci (green arrowhead), their numbers increased significantly after UV or high salt treatment. The UV-experiment was repeated 26 times with similar results. Bars, 10 μm. **b:** fLM of ghost cells from *Anabaena* wild-type and *Anabaena* Cis11–sfGFP. Whereas wild-type ghost cells only showed a faint green autofluorescence, ghost cells from *Anabaena* Cis11–sfGFP additionally exhibited several green foci. All images were normalized to same contrast and brightness values. Bars, 10 μm. **c/d:** Time-lapse fLM of *Anabaena* Cis11–sfGFP cells revealed the induction of ghost cell formation after UV treatment. Red fluorescence (magenta) is caused by chlorophyll in viable cells, whereas the observation of only green fluorescence is an indicator for dead cells. T = 0 min indicates the start of fLM acquisition ~15 min after UV treatment. Dashed boxes indicate magnified views shown in (c). The experiment was repeated 26 times with similar results. Bars, 10 μm. **e:** Brightfield light microscopy and fLM of UV-treated *Anabaena* wild-type cultures that were stained with FM1-43 membrane dye revealed that ghost cells were stable over weeks (time indicated in days after UV treatment). The experiment was repeated twice with similar results. Bar, 10 μm.

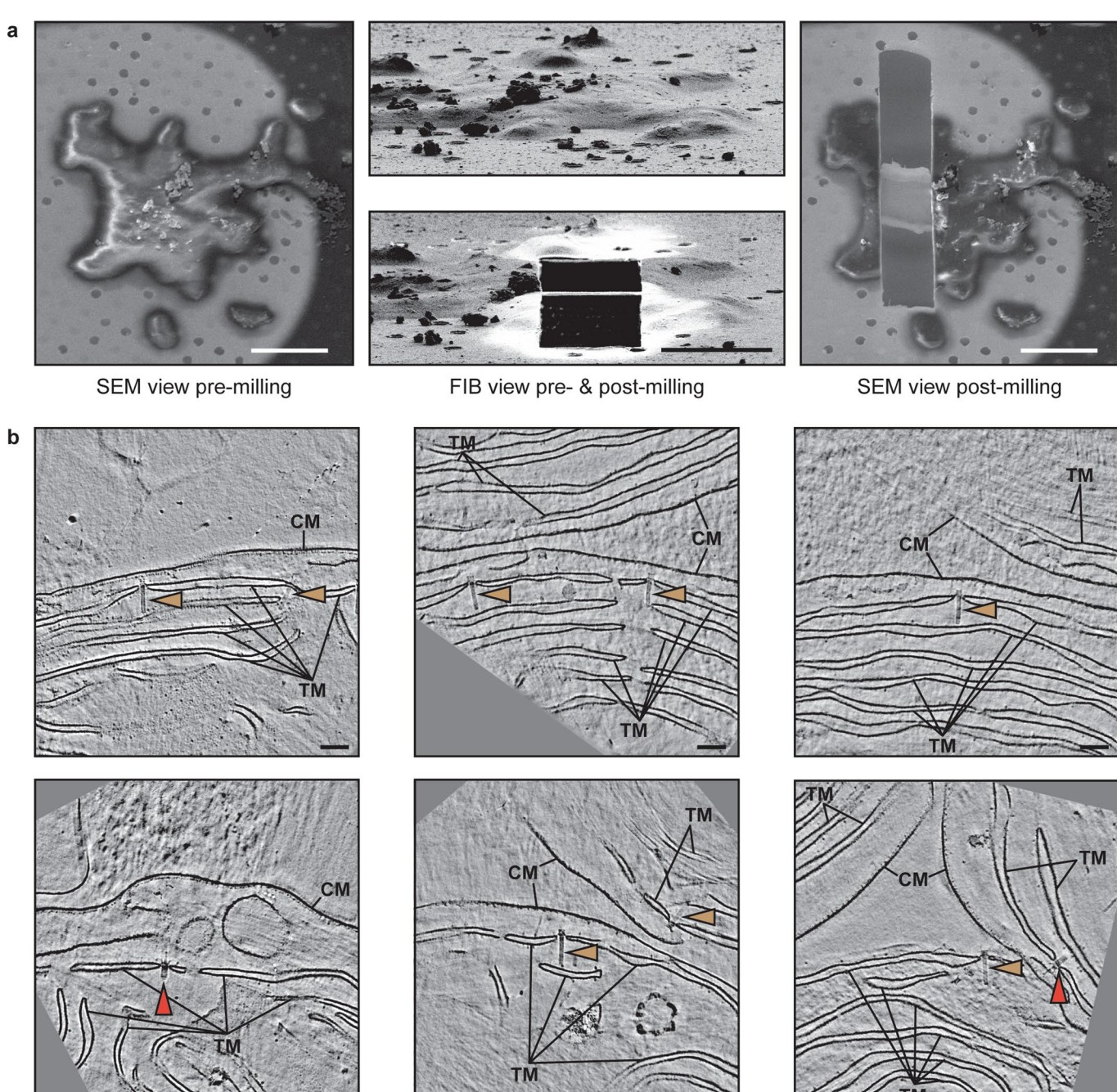

**Extended Data Fig. 8 | CISs are still anchored in the TM within ghost cells. a:** CryoFIB milling of ghost cells. Shown is one example for the preparation of a lamella through a lawn of plunge-frozen ghost cells. Target was identified with a scanning electron microscope (SEM view, pre-milling). Material was removed from top and bottom using the FIB (FIB view, pre- and post-milling). Final lamella was inspected again with SEM (SEM view, post-milling). In total, ten lamellae were prepared on one grid. Bars, 10 μm. **b:** Gallery of slices through ghost cell cryo-tomograms. Outer membrane and peptidoglycan were always absent in all tomograms (n = 72 tomograms). In more than half of the analysed ghost cells (n = 88), the cytoplasmic membrane was no longer intact and in 17 tomograms CISs (brown arrowheads, extended CISs; red arrowheads, contracted CISs) were directly exposed to the environment. CM, cytoplasmic membrane; TM, thylakoid membrane stack. Shown are 17.2 nm thick slices. Similar results were achieved in three independent experiments. Bars, 100 nm.

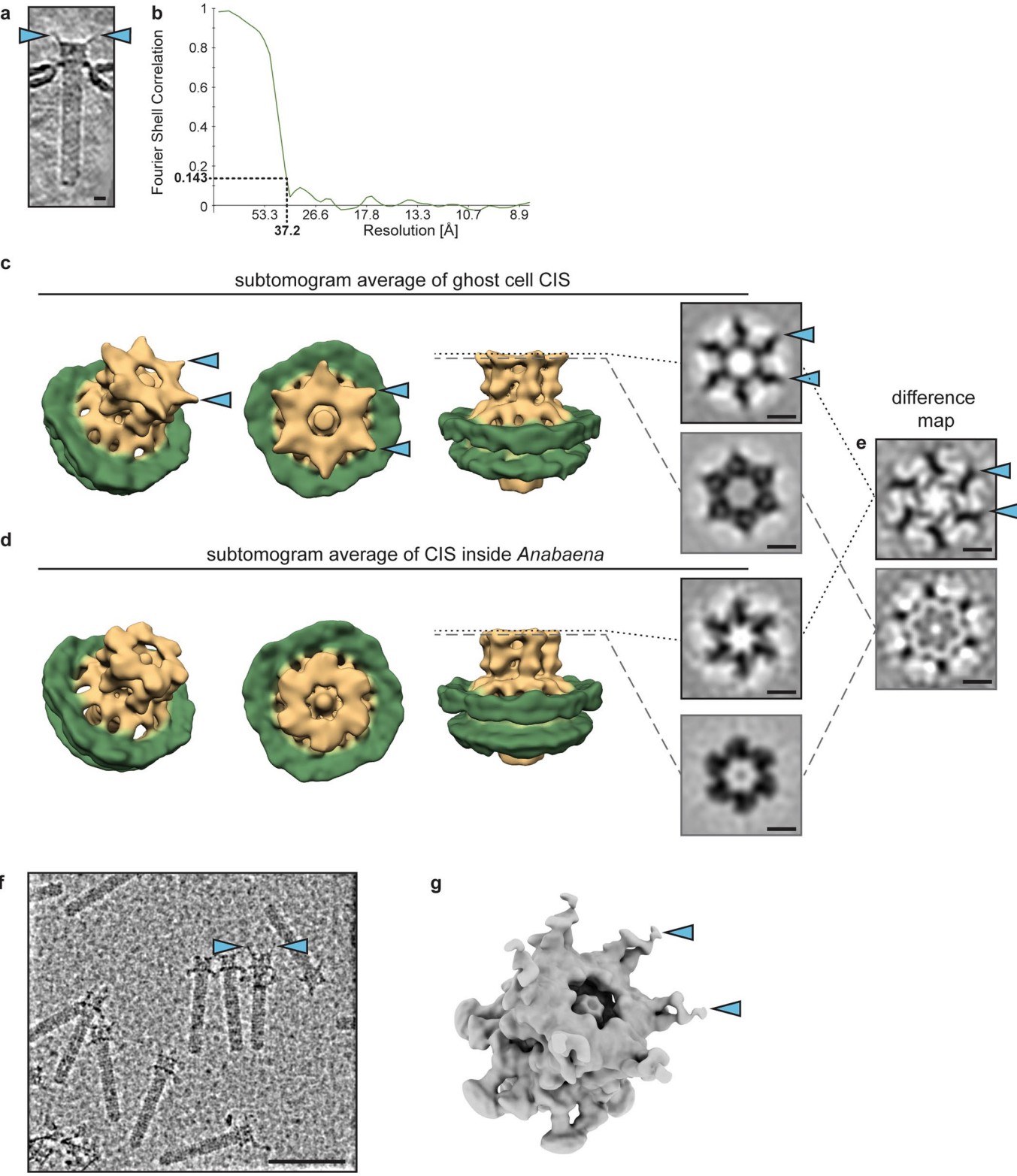

**Extended Data Fig. 9 | CISs in ghost cells harbour protruding tail pins. a:** Another example of CIS in ghost cell showing small tail fibre-like crown protrusions (blue arrowheads; 18 CISs observed in eight individual tomograms). Bar, 10 nm. **b:** Fourier Shell Correlation (FSC) analysis of two half-datasets of subtomogram average of CISs within ghost cells resulted in a resolution of ~37 Å. **c-e:** Isosurface views and perpendicular slices through the crown region (right, height indicated with dotted lines) of subtomogram averages of CISs from intact *Anabaena* filaments (d) and of CISs from ghost cells (c), both filtered to 40 Å resolution. A difference map (e; same height of slices as shown in c/d, right) revealed, that the only major difference were the small tail fibre-like crown protrusions that were only observed in the ghost cell CIS average. See Methods section for details on particle numbers. Bars, 10 nm. **f-g:** Small tail fibre-like protrusions (blue arrowheads) could also be detected in individual 2D micrographs (12 examples over 52 baseplate particles in a subset of 10 micrographs) of purified CISs (f) or in a low-pass filtered SPA map (g). These small tail fibres showed a high flexibility and might correspond to Cis19 domains which could not be modelled. See also Fig. 2e for fitting of atomic model of Cis19 trimer in *in situ* structure. Bar in (f), 100 nm.

**a** cryoET - ghost cell gallery

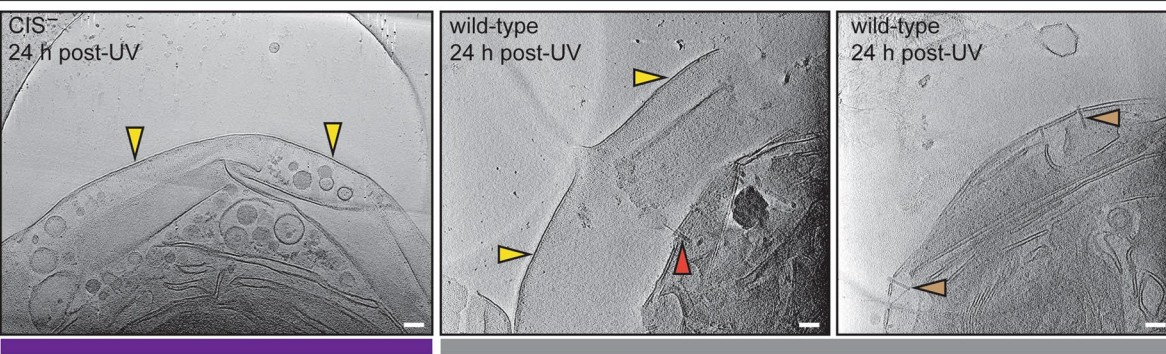

"CM intact" "CM ruptured/absent"

**b**

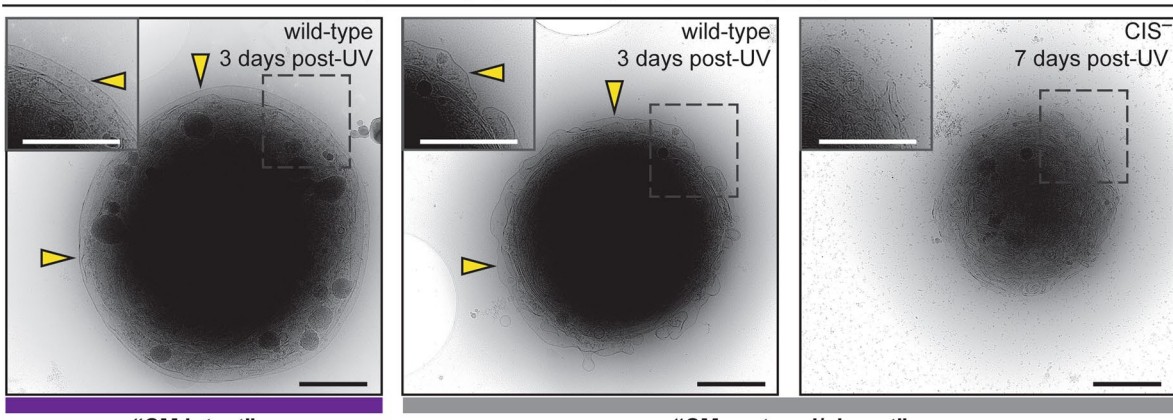

**c** cryoEM 2D projection images - ghost cell gallery

"CM intact" "CM ruptured/absent"

**d**

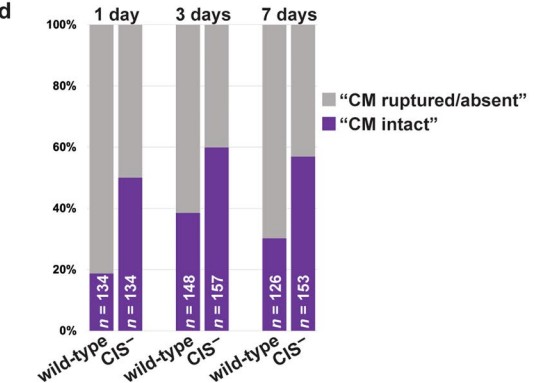

**Extended Data Fig. 10 | See next page for caption.**

**Extended Data Fig. 10 | Analysis of cytoplasmic membrane morphology of ghost cells reveals differences between *Anabaena* wild-type and CIS⁻ mutant. a:** Shown are example cryo-tomograms of ghost cells (shown are 36.6 nm and 44.1 nm thick slices in the left and middle/right panel, respectively). *Anabaena* wild-type and CIS⁻ cultures were plunge frozen 24 h after UV treatment. The resulting tomograms of ghost cells were analysed and classified into two classes according to the morphology of their cytoplasmic membrane (yellow arrowheads) within the field of view of the tomogram: 'CM intact' and 'CM ruptured/absent'. In *Anabaena* wild-type ghost cells, extended (brown arrowhead) and contracted (red arrowhead) CISs were still found being anchored in the outermost TM. Bars, 100 nm. **b:** Shown is the quantification of cytoplasmic membrane morphology of ghost cells in cryo-tomograms (within the field of view). The fraction of ghost cells with intact cytoplasmic membrane was higher in *Anabaena* CIS⁻ (93 %), compared to the wild-type (47 %). *n* indicates the number of analysed tomograms per condition from two independent datasets (see Source Data for detailed numbers). **c:** Shown are example cryoEM 2D projection images of ghost cells (magnified view of dashed grey box in upper left corner). *Anabaena* wild-type and CIS⁻ cultures were plunge frozen 1, 3, and 7 days after UV treatment. Similar to (a/b), the ghost cells were classified into the classes: 'CM intact' and 'CM ruptured/absent'. CM, cytoplasmic membrane (yellow arrowheads). Bars, 1 μm. **d:** Shown is the quantification of cytoplasmic membrane morphology of ghost cells in cryoEM 2D projection images. Compared to the wild-type, ghost cells formed by *Anabaena* CIS⁻ showed a higher fraction of cells with intact cytoplasmic membrane. *n* indicates the number of analysed 2D images per condition from two independent datasets (see Source Data for detailed numbers).

# Reporting Summary

Nature Research wishes to improve the reproducibility of the work that we publish. This form provides structure for consistency and transparency in reporting. For further information on Nature Research policies, see our Editorial Policies and the Editorial Policy Checklist.

## Statistics

For all statistical analyses, confirm that the following items are present in the figure legend, table legend, main text, or Methods section.

| n/a | Confirmed | |
|---|---|---|
| ☐ | ☒ | The exact sample size (*n*) for each experimental group/condition, given as a discrete number and unit of measurement |
| ☐ | ☒ | A statement on whether measurements were taken from distinct samples or whether the same sample was measured repeatedly |
| ☒ | ☐ | The statistical test(s) used AND whether they are one- or two-sided <br> *Only common tests should be described solely by name; describe more complex techniques in the Methods section.* |
| ☒ | ☐ | A description of all covariates tested |
| ☒ | ☐ | A description of any assumptions or corrections, such as tests of normality and adjustment for multiple comparisons |
| ☒ | ☐ | A full description of the statistical parameters including central tendency (e.g. means) or other basic estimates (e.g. regression coefficient) AND variation (e.g. standard deviation) or associated estimates of uncertainty (e.g. confidence intervals) |
| ☒ | ☐ | For null hypothesis testing, the test statistic (e.g. *F*, *t*, *r*) with confidence intervals, effect sizes, degrees of freedom and *P* value noted <br> *Give P values as exact values whenever suitable.* |
| ☒ | ☐ | For Bayesian analysis, information on the choice of priors and Markov chain Monte Carlo settings |
| ☒ | ☐ | For hierarchical and complex designs, identification of the appropriate level for tests and full reporting of outcomes |
| ☒ | ☐ | Estimates of effect sizes (e.g. Cohen's *d*, Pearson's *r*), indicating how they were calculated |

*Our web collection on statistics for biologists contains articles on many of the points above.*

## Software and code

Policy information about availability of computer code

| | |
|---|---|
| Data collection | SPA cryoEM and cryoET data collection: SerialEM 3.7 <br> cryoFIB milling: Thermo Fisher Scientific XT software <br> Light microscopy: Visiview, Zen, LasX, Fiji |
| Data analysis | cryoET and subtomogram averaging: IMOD 4.11, PEET, Dynamo, Chimera <br> single-particle cryoEM: IMOD 4.11, RELION 3.0, crYOLO, COOT, RosettaCM, PHENIX, Chimera, gctf, MolProbity, mtriage, ChimeraX |

For manuscripts utilizing custom algorithms or software that are central to the research but not yet described in published literature, software must be made available to editors and reviewers. We strongly encourage code deposition in a community repository (e.g. GitHub). See the Nature Research guidelines for submitting code & software for further information.

## Data

Policy information about availability of data

All manuscripts must include a data availability statement. This statement should provide the following information, where applicable:
- Accession codes, unique identifiers, or web links for publicly available datasets
- A list of figures that have associated raw data
- A description of any restrictions on data availability

Example tomograms (EMD-13770 - EMD-13772), subtomogram averages (EMD-13773 – EMD-13775) and SPA cryoEM maps (EMD-12029 – EMD-12034, see Table S6) were uploaded to the Electron Microscopy Data Bank. Atomic coordinates of the baseplate (PDB 7B5H) and cap complex (PDB 7B5I) have been uploaded to the Protein Data Bank.

Other datasets used in this study from the Protein Data Bank (PDB): 6J0F, 6RAP, 6J0B, 6RBN, 6J0N, 6RBN, 6RAO

# Field-specific reporting

Please select the one below that is the best fit for your research. If you are not sure, read the appropriate sections before making your selection.

☒ Life sciences ☐ Behavioural & social sciences ☐ Ecological, evolutionary & environmental sciences

For a reference copy of the document with all sections, see nature.com/documents/nr-reporting-summary-flat.pdf

# Life sciences study design

All studies must disclose on these points even when the disclosure is negative.

| | |
|---|---|
| Sample size | All particles (n = 209) found in 99 high-quality tomograms were used for initial in situ subtomogram average of extended CIS. Final particle number after corss-correlation cleaning: 170.<br>All particles (n =78) found in 15 high-quality tomograms were used for initial subtomogram average of CIS in ghost cells. FInal particle number after CC cleaning: 64.<br>All particles (n = 11) found in 6 high-quality tomograms were used for subtomogram average of contracted CIS.<br>For SPA cryoEM, please see Fig. S5 for detailed numbers of particles as well as Methods section.<br><br>For all other experiments, no sample size determination was performed. |
| Data exclusions | Particles which were not fully inside the field of view were excluded. CISs seen in tomograms of bad quality were excluded. |
| Replication | Replication of cryoEM and cryoET findings was not attempted.<br>Replications of light microscopy findings were successful at all attempts and the number of replications are stated in figure legends. |
| Randomization | Extracted particles (for SPA and subtomogram averaging) were randomly assigned to two separate groups to calculate half-maps and gold-standard FSC.<br>For other experiments, no randomization was performed. |
| Blinding | CryoEM projection images of ghost cells released from wild-type and CIS deficient mutant were anonymized and blindly sorted into classes ("CM intact" and "CM ruptured/absent"). For all other experiments, blinding was not attempted. |

# Reporting for specific materials, systems and methods

We require information from authors about some types of materials, experimental systems and methods used in many studies. Here, indicate whether each material, system or method listed is relevant to your study. If you are not sure if a list item applies to your research, read the appropriate section before selecting a response.

## Materials & experimental systems

| n/a | Involved in the study |
|---|---|
| ☐ | ☒ Antibodies |
| ☐ | ☒ Eukaryotic cell lines |
| ☒ | ☐ Palaeontology and archaeology |
| ☐ | ☒ Animals and other organisms |
| ☒ | ☐ Human research participants |
| ☒ | ☐ Clinical data |
| ☒ | ☐ Dual use research of concern |

## Methods

| n/a | Involved in the study |
|---|---|
| ☒ | ☐ ChIP-seq |
| ☒ | ☐ Flow cytometry |
| ☒ | ☐ MRI-based neuroimaging |

## Antibodies

| | |
|---|---|
| Antibodies used | For His-tagged samples, membranes were incubated with 1:5000 HRP-conjugated anti-6xHis antibody (#MA1-21315-HRP, Invitrogen). Alternatively, membranes were incubated with 1:1000 polyclonal rabbit anti-all3324 (anti-Cis1) or anti-all3325 (anti-Cis2) antibody (GenScript) and 1:5000 secondary horseradish peroxidase-conjugated goat anti-rabbit IgG (Abcam). |
| Validation | Validation of antibodies was done by manufacturer (SDS-PAGE of Antigen and Western Blot, Elisa) as well as with western blotting against Anabaena wild-type and CIS deficient mutant. |

## Eukaryotic cell lines

Policy information about cell lines

| | |
|---|---|
| Cell line source(s) | Sf9 insect cells |

| Authentication | None of the used cell lines were authenticated. |
| Mycoplasma contamination | Cell lines were not tested for mycoplasma contamination. |
| Commonly misidentified lines (See ICLAC register) | *Name any commonly misidentified cell lines used in the study and provide a rationale for their use.* |

## Animals and other organisms

Policy information about studies involving animals; ARRIVE guidelines recommended for reporting animal research

| Laboratory animals | Ciliates, Daphniae, Hydroides elegans tubeworm larvae, wax moth larvae |
| Wild animals | No wild animals haven been used in this study. |
| Field-collected samples | Water samples from Lake Zürich were collected at the center of the lake in front of Thalwil, using a twin plankton net (2x 100 µm mesh) tow from a depth of 10 m to the surface. 200 mL of this sample was then stored at 4 °C and used for CIS purification. |
| Ethics oversight | No ethics approval was required. |

Note that full information on the approval of the study protocol must also be provided in the manuscript.

