## [Peer Review File · Nature Microbiology]

Peer Review Information

Journal: Microbiology

Manuscript Title: Structure of a thylakoid-anchored contractile injection system in multicellular cyanobacteria

Corresponding author name(s): Martin Pilhofer

Editorial Notes:

Redactions – transferred manuscripts (mention of the other journal) This manuscript has been previously reviewed at another journal. This document only contains reviewer comments, rebuttal and decision letters for versions considered at Nature **XX**. Mentions of the other journal have been redacted.

Reviewer Comments & Decisions:Decision Letter, initial version:

Dear Martin,

Thank you for submitting your revised manuscript "Structure of a thylakoid-anchored contractile injection system" (NMICROBIOL-21102642). It has now been seen by our referees and their comments are below. The reviewers find that the paper has improved in revision, and therefore we'll be happy in principle to publish it in Nature Microbiology, pending some minor revisions to satisfy the referees' final requests and to comply with our editorial and formatting guidelines.

We will now perform detailed checks on your paper and all associated files, and will send you a checklist detailing our editorial and formatting requirements in about a week. Since the current version of your manuscript is in a PDF format, please email us a copy of the file in an editable format (such as Microsoft Word or LaTeX) so that we can perform these checks-- we can not proceed with PDFs at this stage. Please do not upload the final materials and make any revisions until you receive this checklist from us.

Thank you again for your interest in Nature Microbiology, and please do not hesitate to contact me if you have any questions.

With best regards,

{redacted}

Reviewer #1 (Remarks to the Author):

Speaking as an expert in both photosynthetic organisms and in situ structural biology, I believe this paper by Weiss et al. is truly a landmark study. As stated in my original extensive review, this is a methodological tour-de-force that combines in vitro and in vivo structural techniques in a way that is at the cutting-edge of our field. The biological findings are also novel and quite exciting. The authors go to great lengths to uncover the function of this novel CIS complex, and I feel that further experiments in this direction are beyond the scope of this already comprehensive study.

I supported publication of this paper when I originally reviewed it, and it is even better now after revision. The authors have done an admirable job addressing all of the issues I raised with edits to the manuscript and even substantial new experiments. I have only one suggestion: In the response to the reviewers, the authors make a new finding that cell death along the Anabaena filament is not random but rather frequently occurs next to the original "seed" dead cell (Fig. R1A). They further find that this non-random cell death along the filament is independent of the presence of the CIS. While this could be considered a "negative result", I do think it is very interesting and provides a hint that this thylakoid-embedded CIS may not be involved in neighbor-killing. I would recommend including this result in the paper as a panel in a supplemental figure. But ultimately, I leave this up to the authors' discretion if they want to save this interesting result for follow-up work.

In summary, I fully endorse publication of this work in its current state.

2Re-Reviewed by Benjamin Engel
(Originally Reviewed by Benjamin Engel and Ricardo Righetto)

Reviewer #2 (Remarks to the Author):

This is an exciting piece of work that opens up a whole load of questions about the physiological role of this novel T6SS, and some fascinating questions about interactions between cyanobacteria and other species in their environment. The physiological role still isn't clear, but the structural characterisation is first-class and the authors have made an excellent effort to test some of the obvious possibilities for the physiological role. I think they have made a strong response to the previous reviewers' comments. In fact, in their response to the previous reviewer#1, I don't think they need to be so defensive about the green fluorescence in their ghost cells. It is generally the case that green autofluorescence substantially increases in cyanobacterial ghost cells, although the origin of the extra green fluorescence is not clear. It is NOT the case that they appear greener only because the fluorescence is no longer masked by red fluorescence from the photosynthetic pigments. Judging from Fig. 4 and Fig. S15, that is also the case here. I have some other minor edits to suggest:

1. Still on the green autofluorescence, I think the authors need to make a bit more effort to provide controls to show that the green fluorescence that they claim is from GFP really is from GFP. That applies to Fig. 1b, Fig S2 and Fig S15b. For Fig 1b and Fig S2, no comparison with wt is provided, and for Fig. S15b, the images are too small and low-resolution to be very helpful. Green autofluorescence can sometimes be rather patchy, so some caveats may be needed. Quantitation of mean green fluorescence per cell in wt vs the tagged strains would be helpful.
2. I think the nature of the attachment to the thylakoid membranes deserves a bit more discussion with respect to what we know of thylakoid structure. At first glance, I assumed that the T6SS baseplate spanned the thylakoid lumen (which would entail two transmembrane belts, one to span each membrane either side of the thylakoid sac). However, it's clear from the cartoon in Fig. 5a that this is not what the authors mean. Rather, the baseplate is anchored within a pore in the thylakoid sac. Those pores look to be equivalent to the "fenestrations" or "perforations" which I think were first described by Nevo et al (EMBO J (2007)26:1467-1473). So it would be helpful to use that terminology when discussing the anchoring mechanism. It would be also be helpful if the authors could explain how they rule out the first scenario, I guess based on what they can observe of the lipid bilayers in the neighbourhood of the T6SS.
3. Table S2. For balance, it would be helpful to include a listing of some of those cyanobacterial species that lack T6SS homologs, especially the well-known experimental models.
4. A very minor point, but on p.20 "Bg11 medium" should be "BG11 medium" (the BG stands for Blue-Green).

Reviewer #3 (Remarks to the Author):

The following notes and remarks cover both papers from the Pilhofer lab - by Xu et al. and by Weiss et al. - that described the structure of PVC-like complexes of Algoriphagus and Anabaena, respectively. ^{SEP}

3Despite not having access to the original version of the MSs, it is clear that the authors addressed the comments and critiques of the referees very thoroughly and improved their papers to a point where some minor edits can still be suggested, but they might be superfluous.

Obviously, the function of the two biological systems in question (two different phage tail-like structures) remains unknown. So, this is a major sticking point for both MSs. The description of the structure and illustrations are excellent.

To this CIS expert, the only major problem with both MSs is their Abstracts. The authors have long been proponents of sub-typing the class of contractile injection systems into smaller groups. With these two new papers and new systems (that are nevertheless are very much PCV-like), the authors emphasize in the abstracts how these new systems fit into that classification ("compatible" or "incompatible" with T6SS or eCIS). I do not think this is relevant because the classification is built using a very small dataset and hence can be incorrect. Both abstracts must emphasize what have been found in these papers rather than (dis)agreement with some classification system. The latter would have been acceptable for, and count even constitute the main subject of, a bioinformatic paper. But in this case, the Abstract must be a summary of experimental discoveries.

Some nitpicking.

In the MS "Identification and structure of an extracellular contractile injection system in marine bacteria"

Line 49. Why is Ref. 9 cited? What the process of membrane puncturing studied in that MS? ^{CL}_{SEP}Line 53. Not just "pyocins" (the term came out of nowhere), but R-type pyocins.

Line 220. This paper <https://www.nature.com/articles/415553a> (PMID: 11823865) is the most appropriate reference here.

Decision Letter, final checks:

Dear Martin,

Thank you for your patience as we've prepared the guidelines for final submission of your Nature Microbiology manuscript, "Structure of a thylakoid-anchored contractile injection system" (NMICROBIOL-21102642). Please carefully follow the step-by-step instructions provided in the attached file, and add a response in each row of the table to indicate the changes that you have made. Ensuring that each point is addressed will help to ensure that your revised manuscript can be swiftly handed over to our production team.

If you have not done so already, please alert us to any related manuscripts from your group that are

4under consideration or in press at other journals, or are being written up for submission to other journals (see: <https://www.nature.com/nature-research/editorial-policies/plagiarism#policy-on-duplicate-publication> for details).

In recognition of the time and expertise our reviewers provide to Nature Microbiology's editorial process, we would like to formally acknowledge their contribution to the external peer review of your manuscript entitled "Identification and structure of an extracellular contractile injection system in marine bacteria". For those reviewers who give their assent, we will be publishing their names alongside the published article.

Nature Microbiology offers a Transparent Peer Review option for new original research manuscripts submitted after December 1st, 2019. As part of this initiative, we encourage our authors to support increased transparency into the peer review process by agreeing to have the reviewer comments, author rebuttal letters, and editorial decision letters published as a Supplementary item. When you submit your final files please clearly state in your cover letter whether or not you would like to participate in this initiative. Please note that failure to state your preference will result in delays in accepting your manuscript for publication.

Cover suggestions

As you prepare your final files we encourage you to consider whether you have any images or illustrations that may be appropriate for use on the cover of Nature Microbiology. Covers should be both aesthetically appealing and scientifically relevant, and should be supplied at the best quality available. Due to the prominence of these images, we do not generally select images featuring faces, children, text, graphs, schematic drawings, or collages on our covers. We accept TIFF, JPEG, PNG or PSD file formats (a layered PSD file would be ideal), and the image should be at least 300ppi resolution (preferably 600-1200 ppi), in CMYK colour mode. If your image is selected, we may also use it on the journal website as a banner image, and may need to make artistic alterations to fit our journal style. Please submit your suggestions, clearly labeled, along with your final files. We'll be in touch if more information is needed.

Nature Microbiology has now transitioned to a unified Rights Collection system which will allow our Author Services team to quickly and easily collect the rights and permissions required to publish your work. Approximately 10 days after your paper is formally accepted, you will receive an email in providing you with a link to complete the grant of rights. If your paper is eligible for Open Access, our Author Services team will also be in touch regarding any additional information that may be required to arrange payment for your article. Please note that you will not receive your proofs until the publishing agreement has been received through our system.

Please note that Nature Microbiology is a Transformative Journal (TJ). Authors may publish their research with us through the traditional subscription access route or make their paper immediately open access through payment of an article-processing charge (APC). Authors will not be required to make a final decision about access to their article until it has been accepted. Find out more about Transformative Journals

Authors may need to take specific actions to achieve compliance with funder and institutional open access mandates. For submissions from January 2021, if your research is supported by a funder that requires immediate open access (e.g.according to [Plan S principles](https://www.springernature.com/gp/open-research/plan-s-compliance)) then you should select the gold OA route, and we will direct you to the compliant route where possible. For authors selecting the subscription publication route our standard licensing terms will need to be accepted, including our [self-archiving policies](https://www.springernature.com/gp/open-research/policies/journal-policies). Those standard licensing terms will supersede any other terms that the author or any third party may assert apply to any version of the manuscript.

Please use the following link for uploading all the required materials:

{redacted}

With best regards,

{redacted}

Author Rebuttal, to initial comments:

Reviewers' comments in Black.

Authors' responses in Blue.

Reviewer #1 (Remarks to the Author):

Speaking as an expert in both photosynthetic organisms and in situ structural biology, I believe this paper by Weiss et al. is truly a landmark study. As stated in my original extensive review, this is a methodological tour-de-force that combines in vitro and in vivo structural techniques in a way that is at the cutting-edge of our field. The biological findings are also novel and quite exciting. The authors go to great lengths to uncover the function of this novel CIS complex, and I feel that further experiments in this direction are beyond the scope of this already comprehensive study.

I supported publication of this paper when I originally reviewed it, and it is even better now after revision. The authors have done an admirable job addressing all of the issues I raised with edits to the manuscript and even substantial new experiments. I have only one suggestion: In the response to the reviewers, the authors make a new finding that cell death along the Anabaena filament is not random but rather frequently occurs next to the original "seed" dead cell (Fig. R1A). They further find that this non-random cell death along the filament is independent of the presence of the CIS. While this could be considered a "negative result", I do think it is very interesting and provides a hint that this thylakoid-embedded CIS may not be involved in neighbor-killing. I would recommend including this result in the paper as a panel in a supplemental figure. But ultimately, I leave this up to the authors' discretion if they want to save this interesting result for follow-up work.

In summary, I fully endorse publication of this work in its current state.

6Re-Reviewed by Benjamin Engel
(Originally Reviewed by Benjamin Engel and Ricardo Righetto)

We thank the reviewers for accepting to re-review our manuscript and for the encouraging feedback. As our data suggested that cell death/ghost cell release is independent of the presence of CISs, we prefer to not include the data in the current manuscript.

Reviewer #2 (Remarks to the Author):

This is an exciting piece of work that opens up a whole load of questions about the physiological role of this novel T6SS, and some fascinating questions about interactions between cyanobacteria and other species in their environment. The physiological role still isn't clear, but the structural characterisation is first-class and the authors have made an excellent effort to test some of the obvious possibilities for the physiological role. I think they have made a strong response to the previous reviewers' comments. In fact, in their response to the previous reviewer#1, I don't think they need to be so defensive about the green fluorescence in their ghost cells. It is generally the case that green autofluorescence substantially increases in cyanobacterial ghost cells, although the origin of the extra green fluorescence is not clear. It is NOT the case that they appear greener only because the fluorescence is no longer masked by red fluorescence from the photosynthetic pigments. Judging from Fig. 4 and Fig. S15, that is also the case here. I have some other minor edits to suggest:

We thank the reviewer for taking the time to evaluate our manuscript and for their comment on the observed green autofluorescence.

1. Still on the green autofluorescence, I think the authors need to make a bit more effort to provide controls to show that the green fluorescence that they claim is from GFP really is from GFP. That applies to Fig. 1b, Fig S2 and Fig S15b. For Fig 1b and Fig S2, no comparison with wt is provided, and for Fig. S15b, the images are too small and low-resolution to be very helpful. Green autofluorescence can sometimes be rather patchy, so some caveats may be needed. Quantitation of mean green fluorescence per cell in wt vs the tagged strains would be helpful.

We included an additional supplementary Figure (now Figure S2) with a comparison of fLM images acquired at 488nm excitation wavelength of *Anabaena* wild-type and *Anabaena* expressing Cis11-sfGFP. These images (shown in grey and FIRE lookup table with intensity calibration bars) highlight the fact that GFP foci are exclusively observed in *Anabaena* expressing Cis11-sfGFP.

2. I think the nature of the attachment to the thylakoid membranes deserves a bit more discussion with respect to what we know of thylakoid structure. At first glance, I assumed that the T6SS baseplate spanned the thylakoid

lumen (which would entail two transmembrane belts, one to span each membrane either side of the thylakoid sac). However, it's clear from the cartoon in Fig. 5a that this is not what the authors mean. Rather, the baseplate is anchored within a pore in the thylakoid sac. Those pores look to be equivalent to the "fenestrations" or "perforations" which I think were first described by Nevo et al (EMBO J (2007)26:1467-1473). So it would be helpful to use that terminology when discussing the anchoring mechanism. It would be also be helpful if the authors could explain how they rule out the first scenario, I guess based on what they can observe of the lipid bilayers in the neighbourhood of the T6SS.

To exclude further confusions, we adjusted the manuscript to make it more clear that CIS are anchored within a pore of the thylakoid membrane stack. As these CIS-harboring pores are well defined and of homogenous size, they differ from the previously described "fenestrations" within in thylakoid membranes, which showed a heterogenous appearance. Therefore, we prefer the nomenclature in the current version of the manuscript.

3. Table S2. For balance, it would be helpful to include a listing of some of those cyanobacterial species that lack T6SS homologs, especially the well-known experimental models.

We think it is sufficient to only include a list of strains harboring homologs of the described CIS gene cluster.

4. A very minor point, but on p.20 "Bg11 medium" should be "BG11 medium" (the BG stands for Blue-Green).

We adjusted the manuscript accordingly.

Reviewer #3 (Remarks to the Author):

The following notes and remarks cover both papers from the Pilhofer lab - by Xu et al. and by Weiss et al. - that described the structure of PVC-like complexes of Algoriphagus and Anabaena, respectively.

Despite not having access to the original version of the MSs, it is clear that the authors addressed the comments and critiques of the referees very thoroughly and improved their papers to a point where some minor edits can still be suggested, but they might be superfluous.

Obviously, the function of the two biological systems in question (two different phage tail-like structures) remains unknown. So, this is a major sticking point for both MSs. The description of the structure and illustrations are excellent.

We appreciate the feedback from the reviewer.

To this CIS expert, the only major problem with both MSs is their Abstracts. The authors have long been proponents of sub-typing the class of contractile injection systems into smaller groups. With these two new papers and new systems (that are nevertheless are very much PCV-like), the authors emphasize in the abstracts how these new systems fit into that classification ("compatible" or "incompatible" with T6SS or eCIS). I do not think this is relevant because the classification is built using a very small dataset and hence can be incorrect.

Both abstracts must emphasize what have been found in these papers rather than (dis)agreement with some classification system. The latter would have been acceptable for, and count even constitute the main subject of, a bioinformatic paper. But in this case, the Abstract must be a summary of experimental discoveries.

As suggested by the reviewer, we modified the abstract as well as the main text to better highlight the

experimental discoveries and less the disagreement with previously described CISs' modes of action.

Final Decision Letter:

Dear Martin,

I am very pleased to accept your Article "Structure of a thylakoid-anchored contractile injection system in multicellular cyanobacteria" for publication in Nature Microbiology. Thank you for having chosen to submit your work to us and many congratulations to you and your co-authors.

Acceptance of your manuscript is conditional on all authors' agreement with our publication policies (see <https://www.nature.com/nmicrobiol/editorial-policies>). In particular your manuscript must not be published elsewhere and there must be no announcement of the work to any media outlet until the publication date (the day on which it is uploaded onto our website).

Please note that Nature Microbiology is a Transformative Journal (TJ). Authors may publish their research with us through the traditional subscription access route or make their paper immediately open access through payment of an article-processing charge (APC). Authors will not be required to make a final decision about access to their article until it has been accepted. Find out more

2about Transformative Journals

Authors may need to take specific actions to achieve compliance with funder and institutional open access mandates. For submissions from January 2021, if your research is supported by a funder that requires immediate open access (e.g. according to Plan S principles) then you should select the gold OA route, and we will direct you to the compliant route where possible. For authors selecting the subscription publication route our standard licensing terms will need to be accepted, including our self-archiving policies. Those standard licensing terms will supersede any other terms that the author or any third party may assert apply to any version of the manuscript.

Congratulations once again to you and your co-authors on putting together such a nice paper, I look forward to seeing it published.